# Compute-Optimal Quantization-Aware Training

**Aleksandr Dremov**[*]   **David Grangier**   **Angelos Katharopoulos**   **Awni Hannun**
Apple Inc.
{alexdremov, grangier, a_katharopoulos, awni}@apple.com

## Abstract

Quantization-aware training (QAT) is a leading technique for improving the accuracy of quantized neural networks. Previous work has shown that decomposing training into a full-precision (FP) phase followed by a QAT phase yields superior accuracy compared to QAT alone. However, the optimal allocation of compute between the FP and QAT phases remains unclear. We conduct extensive experiments with various compute budgets, QAT bit widths, and model sizes from 86.0M to 2.2B to investigate how different QAT durations impact final performance. We demonstrate that, contrary to previous findings, the loss-optimal ratio of QAT to FP training increases with the total amount of compute. Moreover, the optimal fraction can be accurately predicted for a wide range of model sizes and quantization widths using the tokens-per-parameter-byte statistic. From experimental data, we derive a loss scaling law that predicts both optimal QAT ratios and final model performance across different QAT/FP compute allocation strategies and QAT bit widths. We use the scaling law to make further predictions, which we verify experimentally, including which QAT bit width is optimal under a given memory constraint and how QAT accuracy with different bit widths compares to full-precision model accuracy. Additionally, we propose a novel cooldown and QAT fusion approach that performs learning rate decay jointly with quantization-aware training, eliminating redundant full-precision model updates and achieving significant compute savings. These findings provide practical insights into efficient QAT planning and enable the training of higher-quality quantized models with the same compute budget.

## 1 Introduction

As Large Language Models (LLMs) grow in size and on-device applications gain traction (Xu et al., 2024), significant attention has been devoted to reducing inference costs via model compression (Frantar et al., 2022; Lin et al., 2024; Ma et al., 2023). One state-of-the-art method is quantization-aware training (QAT) (Chen et al., 2025a; Lin et al., 2024; Liu et al., 2025; Jacob et al., 2018). To adapt the model to the loss of numerical precision, QAT incorporates quantization directly into the model training process. It has been shown that QAT outperforms post-training quantization (PTQ) (Xiao et al., 2023; Banner et al., 2019), where quantization is applied after training is completed. Moreover, Liu et al. (2025) demonstrated that the best accuracy is achieved when a QAT phase follows a full-precision (FP) training phase.

For models designed for on-device use, the QAT stage is an important part of the training process and is usually planned in advance. As model sizes grow and deployment constraints tighten, practitioners face a critical resource allocation problem: **given a fixed compute budget, how should training time be divided between full-precision pretraining and quantization-aware training?** This decision directly impacts both model quality and deployment efficiency, yet existing guidelines assume fixed allocation ratios regardless of scale. As motivation, we note that Kumar et al. (2025) demonstrated that the error introduced by post-training quantization grows with the size of the pretraining data, which can actually make additional pretraining harmful. Intuitively, analogous to PTQ, having a longer full-precision stage should make subsequent QAT more difficult. While Liu et al. (2025) showed that spending 10% of the training steps on QAT is optimal for some setups, the authors did not explore how this proportion varies across different training lengths and model sizes.

---

[*]Work done at Apple during internship while studying at École Polytechnique Fédérale de Lausanne (EPFL), aleksandr.dremov@epfl.ch.

In this work, we show that previous conclusions about optimal QAT length do not hold with an increased compute budget. Through a series of experiments with different model sizes and token counts, we demonstrate that the optimal fraction of QAT compared to the total training length increases with the total compute budget. This optimum can be accurately predicted for a wide range of setups using the **tokens-per-parameter-byte** statistic. Additionally, we propose a loss scaling law as a function of model parameter count ($N$), token count spent on full-precision training ($D_{fp}$), token count spent on QAT ($D_{qat}$), and QAT bit-width ($B$). The fitted law accurately captures the growth of the optimal QAT fraction with compute scale. The key contributions of this study are:

- Unlike previously assumed, we find that the optimal fraction allocated for QAT increases with the growth of the **tokens-per-parameter-byte** statistic. This finding allows higher-quality quantized models to be achieved with the same initial compute budget (figure 1 (**Left**)).

- We propose a loss scaling law that captures the optimal QAT fraction phenomenon and models the final expected loss of the FP and QAT pipeline (figure 1 (**Right**)). We use the scaling law to make further predictions, including which QAT bit-width is optimal under a given memory constraint and how QAT accuracy with different bit-widths compares to full-precision model accuracy.

- We propose a novel approach: **QAT & Learning Rate Cooldown Fusion**—a scheme where learning rate decay is performed jointly with quantization-aware training, eliminating redundant full-precision updates and achieving better accuracy for the same token count.

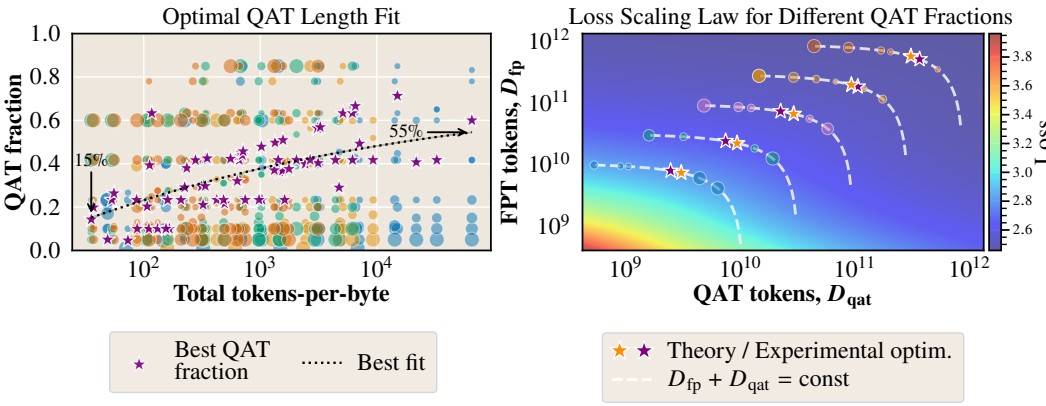

Figure 1: **On the left,** experimental and predicted optimal QAT fractions as a function of tokens-per-parameter-byte are shown. Different colors represent models of varying sizes, while point sizes indicate final perplexity normalized across experiments with identical total token counts for each model size. Results span multiple QAT bit-widths, and optimal QAT fraction values for endpoints are displayed. The plot demonstrates that the optimal QAT fraction increases with the full training tokens-per-parameter-byte statistic. **On the right,** loss scaling law predictions for a 4-bit QAT 396M parameter model across varying QAT and FP training lengths. Both experimental and theoretical optima are shown. The optimal QAT fraction predicted by the loss scaling law for each total token count closely matches the experimentally observed fraction.

## 2 RELATED WORK

**Quantization of LLMs.** Quantization is a method for reducing both the memory footprint and the computational requirements of neural networks by lowering the precision used in the network. By reducing the bit-width of the weights, activations, or both, quantization enables models to run faster, consume less power, and use less memory, which is particularly beneficial for deployment on resource-constrained devices. There are different quantization techniques, including post-training quantization (PTQ) (Xiao et al., 2023; Banner et al., 2019) — methods that transform a model after training has been completed and usually require minimal additional computational usage. Another group of methods is quantization-aware training (QAT) (Chen et al., 2025a; Lin et al., 2024; Liu et al., 2025; Jacob et al., 2018), which quantizes a

model during training, allowing the model to better adapt to precision loss. As quantization operations are non-differentiable, training relies on gradient approximations such as the straight-through estimator (Bengio et al., 2013). In contrast to PTQ, QAT requires more computation, as effectively the full model is trained with added quantization-related operations. In this work, we focus on QAT, as it is the method most commonly used in practice to obtain high-quality quantized models.

**Loss Scaling Laws.** Multiple works have previously addressed the problem of predicting final model loss ($L$) as a function of parameter count ($N$) and consumed tokens ($D$) (Bi et al., 2024; Hoffmann et al., 2022; Kaplan et al., 2020). The Chinchilla (Hoffmann et al., 2022) loss model is one of the most commonly used: $L(N, D) = E + AN^{-\alpha} + CD^{-\beta}$, where $A, C, \alpha, \beta$, and $E$ are fitted parameters. Bi et al. (2024) expand on this idea, fitting accuracy as a function of used non-embedding FLOPs (FLOP estimation of model inference without embedding layer calculations), showing that such an approach works better across different model sizes. Additionally, they show that scaling laws are greatly influenced by data quality.

**QAT Loss Scaling Laws.** Chen et al. (2025b) proposed scaling law modeling specifically for QAT loss, adding a QAT-related penalty to the Chinchilla loss model:

$$L(N, D, G) = \underbrace{E + \frac{A}{N^\alpha} + \frac{C}{D^\beta}}_{\text{Chinchilla loss}} + \underbrace{\frac{k \cdot D^{\gamma_D} \cdot (\log_2 G)^{\gamma_G}}{N^{\gamma_N}}}_{\text{QAT error}}, \qquad (1)$$

where $G$ is the quantization granularity (number of elements in each quantization group), $k, \gamma_D, \gamma_G$, and $\gamma_N$ are fitted parameters, and the Chinchilla loss parameters are fixed from the non-quantized model fit. Therefore, this approach effectively models QAT error relative to the full-precision model for the same token and parameter count. However, formulas are fitted exclusively for each quantization bit width, which complicates analysis of the relationship between different bit widths. Kumar et al. (2025) propose precision-aware scaling laws for training and inference, predicting loss from low-precision training and PTQ or QAT.

While the Kumar et al. (2025); Chen et al. (2025b) laws are useful for understanding the final accuracy of a model trained with QAT from scratch, they overlook the fact that QAT is typically resumed from full-precision training to achieve better accuracy (Liu et al., 2025; Zhou et al., 2025a). Our work addresses this issue and presents a novel loss scaling law that explicitly handles the case when QAT is started from a full-precision checkpoint and works across different bit widths.

## 3 Optimal QAT Compute Allocation

To study how loss changes for different combinations of QAT/FP training length, we train models of different sizes ($N$), different FP stage token counts ($D_{\text{fp}}$), QAT token counts ($D_{\text{qat}}$), and different QAT bit widths ($B$). For the smallest model (86.0M parameters), we conduct experiments from 2.3B to 1.4T total tokens, while for the largest model (759.0M parameters), we conduct experiments from 8.5B to 669.2B total tokens. We focus primarily on 1-, 2-, 4-, and 6-bit quantization widths. Full description of our experimental setup is described in appendix A, and token counts and QAT fractions used are reported in appendix N. We also verify that the obtained post-QAT models maintain reasonable accuracy and present a comparison to full-precision models in appendix P. Specifically, our 4- and 6-bit setups achieve quality close to that of the full-precision model for the same total token count, and the drop in quality for 1- and 2-bit is reasonable.

**The main objective of this study is to determine the optimal QAT fraction $f^*$—the fraction of the token count that should be dedicated to QAT for a given total token count.** This can be formalized as the following minimization problem:

$$D_{\text{qat}}^*(N, D_{\text{total}}, B) = \underset{\substack{D_{\text{qat}} \in \mathbb{N}, \\ D_{\text{qat}} + D_{\text{fp}} = D_{\text{total}}}}{\arg\min} L(N, D_{\text{fp}}, D_{\text{qat}}, B), \quad f^*(N, D_{\text{total}}, B) = \frac{D_{\text{qat}}^*(N, D_{\text{total}}, B)}{D_{\text{total}}},$$

where $L(N, D_{\text{fp}}, D_{\text{qat}}, B)$ is the final loss of the setup with $D_{\text{fp}}$ tokens dedicated to FP training and $D_{\text{qat}}$ tokens dedicated to $B$-bit QAT. Intuitively, $f^*$ expresses a trade-off. On one hand, too few QAT steps do not allow the quantized model to adapt to reduced precision. On the other hand, too many QAT steps (at the expense of fewer FP steps) should also lead to worse loss since QAT is trained with gradient approximations for quantization operators, which introduces biased and noisier gradients. Naturally, a trade-off emerges, suggesting that such $f^*$ should be well-defined.

## 3.1 Predicting the Optimal QAT Fraction

In this section, we focus on fitting the optimal QAT fractions directly. To account for different QAT bit widths used, we introduce the **tokens-per-parameter-byte** statistic. This choice was made based on several observations: larger models are generally easier to quantize, models trained for longer are harder to quantize, and smaller QAT bit widths are harder to quantize as well. While being intuitive from a QAT accuracy perspective, it can also predict the optimal fraction with high precision. Figure 2 provides a comparison between the two approaches. It is clearly seen that using tokens-per-parameter-byte provides an interpretable adjustment, facilitating a better fit.

For optimal QAT fraction prediction, we fit a function of the form

$$\widehat{f}(D_{\text{total}}, N, B) = \frac{\exp\left(\log S_{\text{total}} - \frac{a}{\log S_{\text{total}}}\right)}{S_{\text{total}}},$$

$$S_{\text{total}} = \frac{D_{\text{total}}}{N \cdot \frac{B}{8}},$$

where $\widehat{f}$ is the predicted optimal QAT fraction for the total tokens-per-parameter-byte count $S_{\text{total}}$, and $a$ is a fitted parameter. This function choice was made due to the observed almost linear dependency between $S_{\text{total}}$ and optimal $S_{\text{qat}}$ in log-log coordinates, but with the added constraint that $D_{\text{qat}} \leq D_{\text{total}}$. The optimal fit in our setup yields $a = 6.7297$.

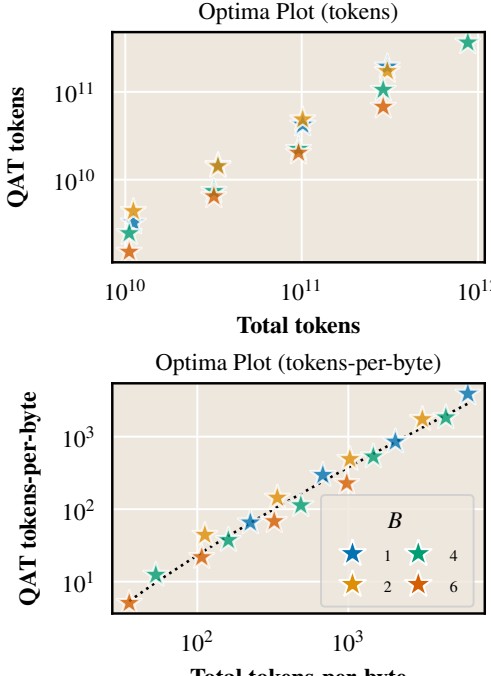

Figure 2: **On the top,** QAT optima for 396M model plotted in **token coordinates**. Different optima for the same total token count and different QAT bit widths can be observed. **On the bottom,** QAT optima for 396M model plotted in **tokens-per-parameter-byte coordinates**. With byte adjustment, different bit widths lie on the proposed fit line better.

**Optimal QAT Fraction Fit Results.** We fit the proposed equation directly using Huber loss (Huber, 1964) and gradient descent optimization. The approach achieves 0.091 MAE in fraction prediction across all model sizes and experiments. We also verify that the error remains low if we remove the largest tested model from the training and evaluate accuracy only on it. The results are displayed in figure 1 **(Left)**. Optimal points lie close to the predicted optimal fraction. We can make the following high-level observations: **the optimal QAT fraction grows faster with $D_{\text{total}}$ for lower bit widths**, **the optimal QAT fraction decreases with model size $N$ increase and fixed $D_{\text{total}}$**. One limitation of the fit is that it is subject to the granularity of the selected experiments—the set of QAT fractions being tested. Also, as we fit only optimal points, we do not use a substantial amount of non-optimal data points, which also contain valuable information about loss behavior. One way to utilize all available data is to model the loss scaling law explicitly and infer optimal QAT fractions from it. We focus on this in section 3.2.

> **Takeaway 1**
>
> Optimal QAT compute allocation fraction is not stationary but grows with total training tokens-per-parameter-byte ($S_{\text{total}}$) and can be predicted from it.

## 3.2 Loss Scaling Law

As described in section 2, Chen et al. (2025b) were able to fit a loss scaling law for QAT started from scratch ($D_{\text{fp}} = 0$). We extend this idea by making the loss scaling law dependent not only on $D_{\text{total}}$ but also on $D_{\text{fp}}$, $D_{\text{qat}}$, and $B$, essentially modeling loss for different QAT fractions and bit-widths. However, we do not follow the same functional form as that proposed by Chen et al. (2025b). This is because in equation 1, the QAT penalty overtakes the Chinchilla loss term at some point with

the growth of token count, which causes the limit of the whole expression to approach infinity as $D_{\text{total}} \to \infty$. This does not align with the expected loss decrease as token count grows and will hinder making any predictions from the scaling law in the future. We propose a loss model in the form of

$$L(N, D_{\text{qat}}, D_{\text{fp}}, B) = \alpha + \underbrace{\frac{\beta}{D_{\text{total}}^{\gamma}} + \frac{\zeta}{N^{\eta}}}_{\text{Chinchilla-like loss}} + \underbrace{\delta(N, D_{\text{qat}}, D_{\text{fp}}, B)}_{\text{QAT fraction-aware penalty}},$$

$$\delta(N, D_{\text{qat}}, D_{\text{fp}}, B) = \underbrace{\theta \cdot 2^{-\kappa \cdot B}}_{\text{Irreducible QAT error}} + \underbrace{\frac{\phi \cdot 2^{-\chi \cdot B}}{N^{\psi} \cdot S_{\text{qat}}^{\omega}}}_{\text{Pure QAT penalty}} + \underbrace{\frac{\lambda \cdot 2^{-\mu \cdot B}}{N^{\nu} \cdot S_{\text{fp}}^{\xi} \cdot S_{\text{qat}}^{\rho}}}_{\text{FP / QAT interaction}},$$

(2)

where all lowercase Greek letters are fitted parameters and $S_{\text{qat}} = \frac{D_{\text{qat}}}{N \cdot \frac{B}{8}}$, $S_{\text{fp}} = \frac{D_{\text{fp}}}{N \cdot \frac{B}{8}}$ are the corresponding tokens-per-parameter-byte. This choice of $\delta(N, D_{\text{qat}}, D_{\text{fp}}, B)$ is motivated by the dependence of the optimal QAT fraction on tokens-per-parameter-byte as discussed in section 3.1; specific motivation for each term is described in equation 2.

**Loss Scaling Law Fit Results.** We fit the proposed equation for **757** total QAT experiments directly using Huber loss (Huber, 1964) and gradient descent optimization—a setup consistent with that of Hoffmann et al. (2022); Chen et al. (2025b). The results are highly dependent on initialization; therefore, we select the best fit out of many random initializations. We achieve similar fit quality across different bit-widths: $R^2 = 0.982$ for 1-bit QAT and $R^2 = 0.991$ for 6-bit QAT, where $R^2$ is the coefficient of determination. Full fit metrics are presented in table 1. We present the fitted formula and its visualizations in appendix D and plot the 3D loss scaling law surface for fixed model size in figure 3. Also, in appendix G we fit formulas independently for each bit-width $B$ as a baseline and verify that the unified formula achieves similar fit metrics. Additional scaling law fit notes are provided in appendix F, appendix K verifies that optimal QAT fraction prediction generalizes to larger model sizes (2.2B), and appendix Q conducts uncertainty analysis and parameters significance tests.

Table 1: Fit metrics for the loss scaling law. We report both the metrics of the loss fit and of the optimal QAT fraction prediction inferred from the loss scaling law. It is seen that the proposed formula in equation 2 provides a good fit of loss as well as of the optimal QAT fraction.

| $B$ | MAE, loss fit | $R^2$, loss fit | MAE, optimal QAT fraction fit |
|---|---|---|---|
| 1 | 0.026 | 0.982 | 0.081 |
| 2 | 0.023 | 0.981 | 0.102 |
| 4 | 0.021 | 0.983 | 0.074 |
| 6 | 0.018 | 0.991 | 0.09 |

The fitted formulas are analytically sound: with an increase of either $D_{\text{fp}}$ or $D_{\text{qat}}$ while the other is fixed, the total loss decreases. Additionally, the proposed form effectively captures the optimal QAT fraction. From experimental results and the fitted loss function, we observe that low-bit QAT is more sensitive to the QAT fraction being optimal. Loss increase for sub-optimal QAT is higher for 1-bit than for 6-bit. Therefore, selecting the optimal QAT fraction is especially important in low-bit settings. To assess the generality of our formulation, we reproduce our findings with different pre-training and QAT hyperparameters and the SlimPajama (Soboleva et al., 2023) dataset in appendix J.

**Takeaway 2**

Final loss after QAT can be accurately predicted from $N$, $D_{\text{qat}}$, $D_{\text{fp}}$, and $B$ for various settings using a single formula. Moreover, the proposed loss scaling law effectively captures the phenomena of optimal QAT fraction and can be used to infer it.

# 4 Loss Scaling Law Implications

The unified loss scaling law obtained in the previous section allows us to analyze practically important QAT properties. In this section, we address previously unanswered questions such as: **"How bad is sub-optimal QAT compute allocation?"**, **"When does QAT match FP accuracy?"**, and **"How should one select QAT precision and parameter count?"**

## 4.1 Evaluating Sub-Optimal QAT Fraction

Using the loss scaling law, we compare the optimal QAT setup with a sub-optimal one. To do this, we calculate **wasted token count**—the number of tokens effectively wasted by a sub-optimal QAT fraction. Figure 4 summarizes wasted token count for different bit widths and token counts; we use 10% QAT as the reference for the sub-optimal setup. Formally, with fitted loss model $L(N, D_{\text{qat}}, D_{\text{fp}}, B)$, we can find such $D^*_{\text{total}}$ and optimal $D^*_{\text{qat}}(N, D^*_{\text{total}}, B)$ that achieve the same loss $l$ as sub-optimal $D^{\text{subopt}}_{\text{qat}}$ and $D^{\text{subopt}}_{\text{total}}$. Specifically:

$$l = L(N, D^{\text{subopt}}_{\text{qat}}, D^{\text{subopt}}_{\text{total}} - D^{\text{subopt}}_{\text{qat}}, B),$$

$$D^*_{\text{total}} = \argmin_{\substack{D'_{\text{total}} \in \mathbb{N} \\ D'_{\text{qat}} = D^*_{\text{qat}}(N, D'_{\text{total}}, B)}} \left| L(N, D'_{\text{qat}}, D'_{\text{total}} - D'_{\text{qat}}, B) - l \right|,$$

$$D_{\text{wasted}} = D^{\text{subopt}}_{\text{total}} - D^*_{\text{total}},$$

and the reported percentage in figure 4 is the fraction of total tokens: $\frac{D_{\text{wasted}}}{D^{\text{subopt}}_{\text{total}}}$.

Two factors influence the wasted tokens magnitude: closeness of 10% to the optimal QAT fraction and the overall flatness of the loss scaling law for high token counts. If the predicted loss is generally flat for some token count, then even high deviation from optimality will yield a minor wasted token count. In the extreme case, for 1-bit QAT, **the same loss can be achieved with just around 50% of compute** if the optimal QAT fraction is used. This effect is still present for 2–4-bit QAT but becomes relatively small for 6-bit.

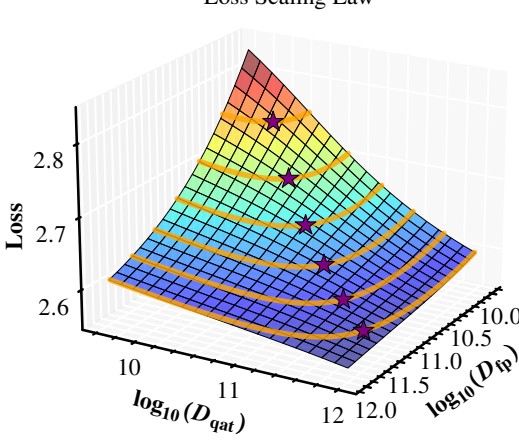

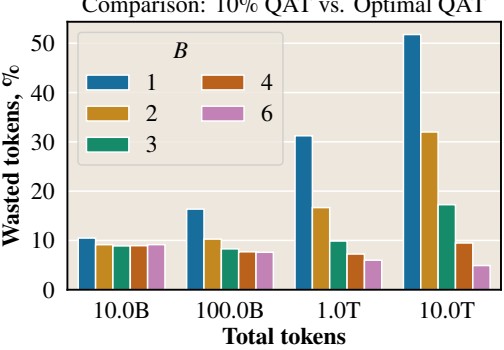

Figure 3: Visualization of the fitted loss scaling law for a 759M model, 1-bit QAT, and different $D_{\text{qat}}, D_{\text{fp}}$. Orange lines represent constant $D_{\text{total}} = D_{\text{qat}} + D_{\text{fp}}$ levels, and stars represent loss minima for each such level. It is clearly seen that the loss structure yields an optimal QAT fraction for a specific $D_{\text{total}}$. The overall phenomenon is consistent with what was discussed in section 3.1.

Figure 4: Comparison of sub-optimal QAT setup with fixed 10% QAT fraction and optimal QAT setup for 1B parameter model. Wasted token count is the number of tokens effectively wasted by not utilizing an optimal QAT fraction setup. That is, if the wasted token count is $n\%$, then the same loss can be achieved with $(100-n)\%$ tokens and optimal QAT fraction. While results vary for different bit widths, the general relationship is similar, revealing high potential savings.

**Takeaway 3**

Suboptimal QAT compute allocation significantly impacts final model performance, especially in low-bit settings. In the extreme case, for 1-bit QAT, **the same loss can be achieved with just around 50% of compute** if the optimal QAT fraction is used.

**Note on Compute Budget–Token Budget Duality.** So far, we have considered token count to be identical to compute budget as compute scales linearly with training token count. However, one may argue that since QAT employs additional operations, its complexity is higher than FP training. As QAT overhead depends only on model size, it becomes negligible with sufficiently large batch size and sequence length (appendix L). Still, in setups where QAT overhead is substantial, one can obtain compute-based optimal QAT fraction from the token-based approach by making a substitution $D_{\text{qat}} = \frac{1}{r} \cdot D'_{\text{qat}}$, where $r > 1$ is the QAT overhead factor in the specific setup and $D'_{\text{qat}}$ is the overhead-aware QAT token count. This will make QAT steps "more expensive" from a loss minimization perspective. In this setup, $D_{\text{fp}} + D'_{\text{qat}} = const$ represents not iso-token levels, but rather iso-FLOP levels. Therefore, the inferred optimal QAT fraction will be adjusted to account for the overhead.

### 4.2 When Does QAT Match FP Accuracy?

We plot the difference in perplexities between QAT and FP models for each total token count. Appendix F describes how we obtain the full-precision model loss scaling law. In summary, we incorporate full-precision training results into the fit with $B = 16$, which not only allows us to predict full-precision model performance but also serves as a regularization for the fit. Figure 5 presents such a plot for models of two different sizes.

**FP Accuracy Reproduction.** The practical observation that larger-parameter models can tolerate lower-bit QAT is clearly observed. A second perspective from which to consider figure 5 is that of optimal QAT bit width. Specifically, for a given total token count, there exists a minimum bit width that matches FP model loss. Therefore, there is no incentive to train higher-bit QAT, as this will not result in better accuracy but only in higher memory usage.

**Takeaway 4**

The proposed loss scaling law effectively captures the empirical observation that larger-parameter models can tolerate lower-bit QAT. Moreover, using the loss scaling law, one can predict a range of total token counts for which QAT accuracy will not differ significantly from that of a FP model.

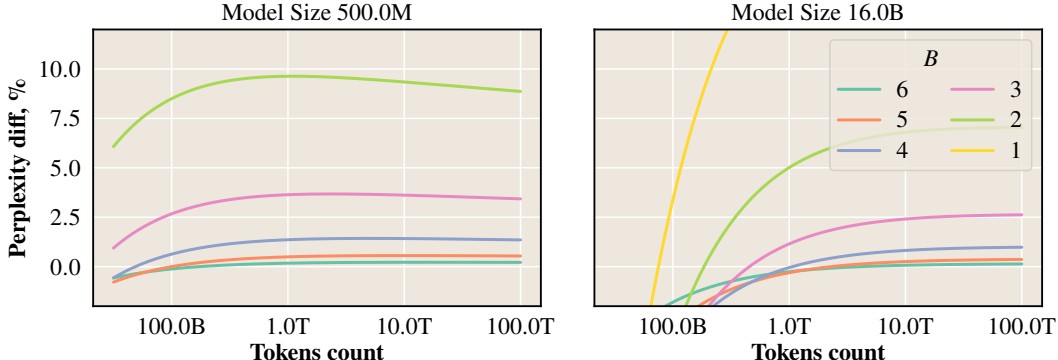

Figure 5: Difference in perplexity between FP loss scaling law and QAT loss scaling law for two model sizes. For QAT, the loss corresponding to the optimal QAT fraction is used. Values below 0 correspond to QAT performing better than the FP model. It is clearly observed that the ability of QAT to match FP loss is greatly influenced by model size and token count. In particular, larger models are able to tolerate lower QAT precision for higher total token count budgets. Additional plot information is present in appendix H.

### 4.3 Parameter–Precision Trade-Off

An interesting question to analyze is **"for a fixed model memory requirement, how should one select QAT precision and parameter count?"**. That is, to fit a specified memory constraint, one can choose high precision at the expense of a lower parameter count or vice versa. This question is practically important as LLM inference is largely bottlenecked by memory bandwidth (Davies et al., 2025; Recasens et al., 2025; Dao et al., 2022). We can derive such optimality from the fitted loss scaling law. The results are presented in figure 6. It is clearly seen that for a fixed memory budget, optimal QAT precision decreases with training FLOP growth. This suggests that for achieving an optimal quantized model within some memory and training compute budget, one should select the parameter count in advance accordingly. We believe this finding to be important for practitioners trying to achieve the best-accuracy model within memory constraints. Figure 6 is verified experimentally in appendix I.

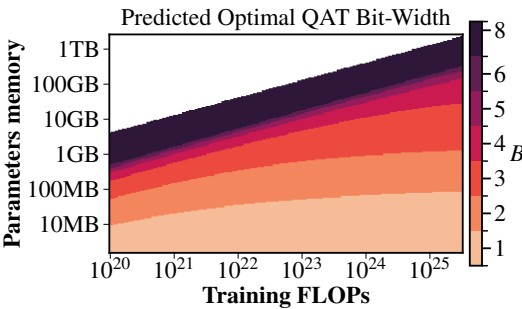

Figure 6: Optimal QAT bit width for different memory budgets and total training budgets. We use the loss corresponding to the optimal QAT fraction. For training FLOPs, we use the estimation $C \sim 6ND$. The white area corresponds to $D < N$, which is not practically important.

---

**Takeaway 5**

The proposed loss scaling law can predict what QAT bit width is optimal for a fixed training compute budget and model memory footprint. It is revealed that for a fixed memory budget, optimal QAT precision decreases with training FLOP growth.

---

## 5 Beyond QAT Compute Fraction: QAT & Cooldown Fusion

Section 3 revealed the importance of advance planning for QAT, accounting for the optimal fraction. This is possible only when one controls the entire pretraining process: both QAT and FP. In this context, it may be worth adjusting the training procedure to make QAT more efficient. Specifically, in this section, we focus on modifications to learning rate scheduling techniques. Currently, a classic way of training models is to perform full FP training with learning rate cooldown, and then start QAT with learning rate re-warmup. We used such a setup for the scaling law in section 3 as it is universally adopted. However, a more optimal scheme may exist.

**QAT & Learning Rate Cooldown Fusion.** Wen et al. (2024) show that re-initializing WSD from a post-cooldown checkpoint rather than from a constant stage yields better results. However, we believe the behavior might be different when resuming training from a checkpoint with QAT. We propose a novel idea: **QAT & Learning Rate Cooldown Fusion**. Motivated by the idea that learning rate cooldown performs low-magnitude adjustments to weights, we speculate that a substantial part of updates during learning rate cooldown gets destroyed by QAT initialization, which, in essence, discards high-precision information. Therefore, we analyze a setup where QAT is started directly from the learning rate constant stage and learning rate cooldown is performed jointly with QAT. A schematic representation of the two schemes is presented in figure 7.

We ran experiments with different model sizes and 4-bit QAT using the described "QAT & Cooldown Fusion" scheme, taking experiments with the classic QAT scheme and optimal QAT fraction as baselines. The results are shown in table 2. In addition to perplexity, we report loss change in "wasted tokens" units. This is the total token distance between corresponding loss points in the scaling law for an optimal QAT fraction. Such a metric is reported for better impact understanding, as small perplexity differences are harder to achieve with high overall token counts. We achieved improvements across all model sizes for 4- and 6-bit widths and all token counts. Results differed for 1- and 2-bit settings; we believe this is due to the large optimal QAT fraction, which makes the

FP fraction small and, consequently, the impact of QAT & cooldown fusion lower. Full results are available in appendix M. While perplexity differences may seem small, judging the difference from the perspective of token count difference is significant. This implies substantial improvements in terms of training cost.

> **Takeaway 6**
>
> The proposed method of learning rate scheduling, "QAT & Learning Rate Cooldown Fusion," further improves QAT efficiency by adjusting the training procedure beyond changing the QAT fraction. While being practically useful, this method also suggests that modifications to the universally accepted QAT pipeline can further improve QAT efficiency.

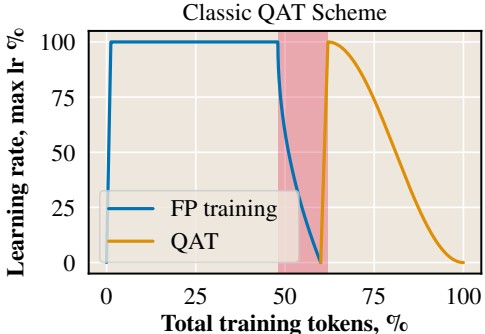 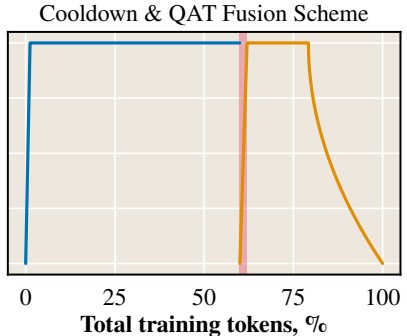

Figure 7: Comparison between two different QAT schemes. In both setups, the QAT fraction is 40%. Red-shaded areas indicate zones with lowered learning rate, which we expect to correspond to minor weight updates that get effectively ignored by QAT initialization. **On the left,** classic QAT scheme visualization: QAT follows fully completed FP training that ends with 20% (of FP training length) learning rate decay. For QAT, the learning rate follows a cosine shape with 5% re-warmup phase. **On the right,** the "QAT & Learning Rate Cooldown Fusion" scheme is displayed. QAT starts directly from the constant learning rate stage with small re-warmup, effectively resuming the FP learning rate scheduler as if QAT was not present at all. QAT ends with 20% cooldown (of total training length). As QAT follows the classic FP learning rate recipe with usual cooldown, we call this approach **QAT & Learning Rate Cooldown Fusion**.

Table 2: Accuracy comparison between the classic QAT scheme and the "QAT & Learning Rate Cooldown Fusion" training scheme. The loss difference is reported in "wasted tokens"—the difference in total token count between optimal QAT fraction loss points in the loss scaling law (formally defined in appendix O). Substantial improvements are noticeable across different model sizes and token counts.

| | | | Perplexity | | Wasted tokens, ↑ |
|---|---|---|---|---|---|
| | | | **Unfused (baseline)** | **Fused (ours)** | **Unfused total tokens, %** |
| **B** | **Model size, M** | $D_{\text{total}}$ | | | |
| 4 | 74 | 1.4T | 16.26 | **16.25**$_{-0.06\%}$ | 2.2% |
| | 163 | 901.3B | 13.51 | **13.49**$_{-0.15\%}$ | 9.2% |
| | 425 | 10.5B | 16.3 | **16.02**$_{-1.72\%}$ | 9.6% |
| | | 31.8B | 13.9 | **13.76**$_{-1.01\%}$ | 10.4% |
| | | 96.0B | 12.62 | **12.54**$_{-0.63\%}$ | 13.6% |
| | 816 | 281.9B | 11.07 | **11.02**$_{-0.45\%}$ | 13.2% |

## 6 CONCLUSION

This work addresses a resource allocation problem in quantization-aware training: how to optimally divide compute between full-precision pretraining and quantization-aware training. Through extensive experiments across model sizes, compute budgets, and quantization bit widths, we challenge existing assumptions and provide practical guidelines for efficient QAT planning. Our key contributions are:

- **Discovery of Compute-Dependent Optimal QAT Fractions.** Through extensive experiments across different model sizes, compute budgets, and QAT bit widths, we demonstrate that previous assumptions about optimal QAT allocation do not hold as compute budgets increase. Our findings reveal that the optimal QAT fraction is not a fixed percentage but rather increases with the total compute budget, specifically with the tokens-per-parameter-byte statistic. This challenges the previous conclusion that 10% is universally optimal for QAT length relative to total training length. We demonstrate that using suboptimal QAT fractions can result in substantial compute waste, with extreme cases showing that the same loss can be achieved with just around 50% of the compute when optimal QAT fractions are used, particularly for low-bit quantization scenarios.

- **Comprehensive Loss Scaling Law.** We derive a comprehensive loss scaling law that models the final expected loss of the full-precision and quantization-aware training pipeline as a function of QAT bit width, model parameter count, and token counts for both training phases. This scaling law not only captures the optimal QAT fraction phenomenon but also enables prediction of final model loss across different QAT/FP allocation strategies. From the scaling law, we infer which QAT bit width is optimal under a given memory constraint and how QAT accuracy compares to FP model accuracy.

- **Cooldown and QAT Fusion Technique.** We introduce a novel approach that performs learning rate decay jointly with quantization-aware training, eliminating redundant full-precision updates and achieving significant compute savings. While being practically useful, this method also suggests that modifications to the universally accepted QAT pipeline can further improve QAT efficiency.

**Limitations.** While we checked the generality of our formulation by performing experiments with a different dataset and hyperparameters in appendix J, our work still focuses on a specific LLM architecture, and exact results may differ for different model types. However, we expect the overall observed phenomena to be consistent across different architectures and datasets.

**Future Work.** We identify several research directions worth exploring. First, the relationship between the optimal QAT fraction and pretraining precision remains unknown. This direction is especially interesting with the emergence of 8-bit floating-point training (Peng et al., 2023) and even 4-bit training (Zhou et al., 2025b; Wang et al., 2025). Second, we are interested in how the observed phenomena are preserved across different training stages. Specifically, how does the optimal QAT fraction change when the full-precision training stage incorporates additional stages such as Supervised Fine-Tuning (SFT) (Ouyang et al., 2022), Reinforcement Learning (RL) (Rafailov et al., 2023; Schulman et al., 2017), or multimodal training? We speculate on these questions in appendix R.

**Reproducibility Statement.** We report exact hyperparameters and training approaches used in appendix A. Additional experimental information that should facilitate reproduction is summarized in appendix B, C, and N.

**Ethics Statement: LLM Use Disclosure.** LLMs such as Anthropic (2025); Mistral AI (2025) were used in the preparation of this paper exclusively for improving grammar and wording.

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

## A  TRAINING SETUP

We use a decoder-only transformer (Vaswani et al., 2017) identical to Llama 2 (Touvron et al., 2023). The architecture incorporates SwiGLU activations (Shazeer, 2020), RoPE (Su et al., 2024), RMSNorm (Zhang & Sennrich, 2019), alternating attention and feed-forward layers, and tied embedding and language-modeling head weights. We use the Adam optimizer ($\beta_1 = 0.9, \beta_2 = 0.99, \varepsilon = 10^{-8}$)

with decoupled weight decay of 0.01 (Loshchilov & Hutter, 2017; Kingma & Ba, 2015) for all parameters outside the embedding and normalization layers. All experiments are trained with bfloat16 automatic mixed precision (Micikevicius et al., 2018). Training is conducted on the DCLM dataset (Li et al., 2024), tokenized with the Llama 2 tokenizer with a 32,000-token vocabulary. We merge all tokenized documents into a single sequence with appropriate delimiting tokens and take chunks of 1024 tokens (used sequence length) for the batch—an approach also known as "concat-and-chunk" (Pouransari et al., 2024). The dataset is split into training and validation sets, and validation perplexity is used for evaluation. For QAT algorithms, we rely on ParetoQ (Liu et al., 2025) for our setups, as this method achieves state-of-the-art accuracy across different bit widths by combining different approaches. The following subsections provide in-depth descriptions of each part of the training pipeline.

## A.1 FULL-PRECISION TRAINING

The choice of learning-rate scheduler is an important aspect of our work. While cosine learning-rate scheduling is widely used, achieving optimal model loss for a specific token count requires matching the training duration to the cosine scheduler length (Hoffmann et al., 2022). To obtain comparable experiments, we would need to train models from scratch for each specific final token count, which is computationally wasteful. Therefore, we train full-precision models with the warmup–stable–decay (WSD) learning-rate scheduler (Hägele et al., 2024; Hu et al., 2024). The main advantage of WSD is the ability to obtain models for any desired total token count without needing to train from scratch; this can be achieved by resuming training from the constant-stage checkpoint and performing a learning-rate cooldown to achieve the needed token count. Hägele et al. (2024) showed that WSD accuracy closely follows that of cosine, making it an optimal choice for our setup, where many checkpoints for different token counts are needed. In our experiments, we follow the optimal setup from Hägele et al. (2024); Dremov et al. (2025): we perform 1,000 steps of warmup and a 20% cooldown stage with a *1 - sqrt* learning-rate cooldown shape.

For different model parameter counts, we vary the number of layers and hidden dimensions, using Hoffmann et al. (2022) as a reference. Our configurations and parameter counts are reported in appendix B. For learning rate and batch size selection, we follow the scaling law proposed by Bi et al. (2024). We choose the optimal batch size and learning rate corresponding to the average token count of the conducted experiments for each model size. Since the achieved loss is stable for wide ranges around the optimal batch size and learning rate (Bi et al., 2024; Zhou et al., 2025a), we remain close to the optimal learning hyperparameters for all our experiments. We report our settings for each model size in appendix C.

## A.2 QUANTIZATION

We rely on ParetoQ (Liu et al., 2025) for our quantization setups, as this approach achieves state-of-the-art accuracy across bit widths. Specifically, we use different algorithms for different bit widths: Elastic Binarization (Liu et al., 2022) for 1-bit quantization; LSQ (Esser et al., 2020) for 3-bit and higher quantization; and SEQ for 2-bit quantization (Liu et al., 2025). Additionally, this approach makes our results generalizable to different QAT algorithms. Each setup employs per-output-feature quantization scales. While it is common not to quantize embeddings and the language modeling head (LM head), the ParetoQ approach shows negligible accuracy drop when quantizing embeddings and the LM head to 4 bits. Since embeddings constitute a substantial portion of parameters for small models, we quantize embeddings as well, but not to fewer than 4 bits in all setups. That is, we quantize embeddings to 6 bits for 6-bit QAT, but to 4 bits for 4-, 2-, and 1-bit QAT experiments.

## A.3 QAT

For QAT, we follow the same setup as for full-precision training, except for the learning rate schedule. At the start of QAT, we restore data readers from the full-precision checkpoint, which makes QAT and FP training data mutually exclusive for each experiment. Since we do not need QAT checkpoints at different token counts, we use cosine learning rate decay with 5% warmup and decay the learning rate to zero. The quantized model is initialized from an appropriate post-cooldown full-precision model, with quantization scale initialization as described by Esser et al. (2020); Liu et al. (2025; 2022) (appendix A.4). We disable weight decay for quantization scales.

## A.4 QAT Algorithms

As described in appendix A.2, we use different quantization algorithms for different $B$. In this section, we summarize them for the reader's convenience.

Typically, QAT algorithms employ a version of the uniform quantization function:

$$\widehat{W_R^i} = \lfloor \frac{W_R^i - \beta}{\alpha} \rceil,$$

$$W_Q^i = \alpha \widehat{W_R^i} + \beta,$$

where $W_R$ is the original floating-point-valued weight, $\widehat{W_R}$ is the quantized integer-valued weight, $W_Q$ is the quantized-dequantized floating-point-valued weight, and $\alpha, \beta$ are parameters specific to the $i$-th quantization group. In our work, we use per-output-feature quantization groups. During training, $W_Q$ is used to conduct calculations, and during inference, the model is stored as integer-valued weights $\widehat{W_R}$. Below, we present details about the different algorithms we used.

**Elastic Binarization (1-bit).** Liu et al. (2022; 2025) propose such a quantization scheme for $\widehat{W_R}$ taking values from $\{-1, 1\}$:

$$\widehat{W_R}^i = \text{Sign}(W_R^i),$$

$$W_Q^i = \alpha \widehat{W_R}^i,$$

where initially $\alpha = \frac{\|W_R^i\|_{l_1}}{n_{W_R^i}}$, and such straight-through (Bengio et al., 2013) estimator gradient estimations are used:

$$\frac{\partial W_Q^i}{\partial W_R^i} \approx 1_{|\frac{w_R^i}{\alpha}| < 1},$$

$$\frac{\partial W_Q^i}{\partial \alpha} \approx \text{Sign}(W_R^i).$$

**Stretched Elastic Quantization (2-bit).** Liu et al. (2025) propose the following quantization scheme for 2-bit $\widehat{W_R}$:

$$\widehat{W_R}^i = \lfloor \text{Clip}(\frac{W_R^i}{\alpha}, -1, 1) \times 2 - \frac{1}{2} \rceil,$$

$$W_Q^i = \frac{\alpha}{2}(\widehat{W_R}^i + \frac{1}{2}),$$

where initially $\alpha = \max(|W_R^i|)$, and such gradient estimations are used:

$$\frac{\partial W_Q^i}{\partial W_R^i} \approx 1_{|\frac{w_R^i}{\alpha}| < 1},$$

$$\frac{\partial W_Q^i}{\partial \alpha} \approx \widehat{W_R^i} - \frac{W_R^i}{\alpha} \cdot 1_{|\frac{w_R^i}{\alpha}| < 1}.$$

**Learned Step Size Quantization (3-bit and Higher).** Esser et al. (2020) propose the following quantization scheme for $W_Q$, which is a standard quantization scheme with $\beta = 0$:

$$\widehat{W_R}^i = \lfloor \text{Clip}(\frac{W_R^i}{\alpha}, -2^{B-1}, 2^{B-1} - 1) \rceil,$$

$$W_Q^i = \alpha \widehat{W_R}^i,$$

where initially $\alpha = \max(|W_R^i|)$, and such gradient estimations are used:

$$\frac{\partial W_Q^i}{\partial W_R^i} \approx 1_{-2^{B-1} < \frac{w_R^i}{\alpha} < 2^{B-1} - 1},$$

$$\frac{\partial W_Q^i}{\partial \alpha} \approx \widehat{W_R}^i - \frac{W_R^i}{\alpha} \cdot 1_{-2^{B-1} < \frac{w_R^i}{\alpha} < 2^{B-1} - 1}.$$

## B    MODEL CONFIGURATIONS

Table 3 summarizes the different transformer model configurations used. As noted, we use the number of layers and hidden dimensions from the configurations table of Hoffmann et al. (2022).

Table 3: Transformer hyperparameters used across experiments. Parameter counts are also reported.

| $d_{\text{model}}$ | $\text{ffn}_{\text{size}}$ | $\text{kv}_{\text{size}}$ | $n_{\text{heads}}$ | $n_{\text{layers}}$ | $N$ (M) | $N_{\text{no emb}}$ (M) |
|---|---|---|---|---|---|---|
| 640 | 2,560 | 64 | 10 | 10 | 86 | 65 |
| 768 | 3,072 | 64 | 12 | 18 | 194 | 169 |
| 1,280 | 5,120 | 128 | 10 | 18 | 396 | 355 |
| 1,536 | 6,144 | 128 | 12 | 25 | 759 | 709 |
| 2,176 | 8,704 | 128 | 17 | 28 | 2,191 | 2,121 |

## C    TRAINING HYPERPARAMETERS

As noted in appendix A.1, for learning rate and batch size selection, we follow the scaling law proposed by Bi et al. (2024). Table 4 describes the chosen hyperparameters for each model size.

Table 4: Main hyperparameters used during training. Learning rate and batch size selection follow those of Bi et al. (2024).

| Model size (M) | Learning rate | Global batch size (tokens) |
|---|---|---|
| 86 | 9.54e-04 | 1,097,728 |
| 194 | 8.93e-04 | 1,302,528 |
| 396 | 7.33e-04 | 1,572,864 |
| 759 | 7.29e-04 | 2,129,920 |
| 2,191 | 6.72e-04 | 2,490,368 |

## D Fitted Loss Scaling Law Formula

In figure 8, we present the loss scaling law fitted to all our experiments. For simplicity, we substitute: $S_{\text{qat}} = \frac{D_{\text{qat}}}{N \cdot \frac{B}{8}}$, $S_{\text{fp}} = \frac{D_{\text{fp}}}{N \cdot \frac{B}{8}}$. Additionally, we plot experimental data and loss scaling law heatmaps in figure 10 and optimal QAT fraction predictions in the same format as figure 1 (**Left**) in figure 9.

$$L(N, D_{\text{qat}}, D_{\text{fp}}, B) = 1.598 + \frac{2477.0}{D_{\text{total}}^{0.4089}} + \frac{57.64}{N^{0.2148}} + 0.4297 \cdot 2^{-1.41 \cdot B} + \frac{1091.0 \cdot 2^{-1.212 \cdot B}}{N^{0.4004} \cdot S_{\text{qat}}^{0.076}} +$$

$$\frac{138.8 \cdot 2^{-0.0833 \cdot B}}{N^{0.2135} \cdot S_{\text{fp}}^{0.4819} \cdot S_{\text{qat}}^{0.1903}}$$

Figure 8: Fitted loss scaling law formula. This is a unified scaling law that predicts QAT loss for various $N$, $D_{\text{qat}}$, $D_{\text{fp}}$, and $B$.

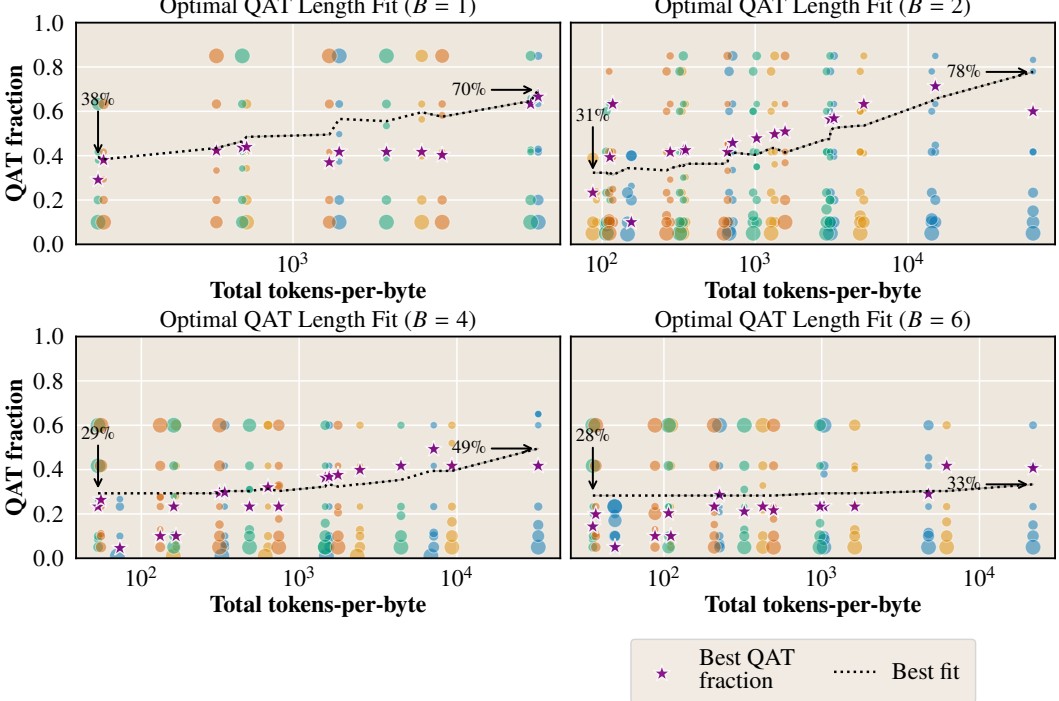

Figure 9: Optimal QAT fraction predictions inferred from the loss scaling law (section 3.2). Note that figure 1 uses a formula from section 3.1, which is less precise but allows a simple line prediction across all bit-widths and model sizes as it depends only on $S_{\text{total}}$.

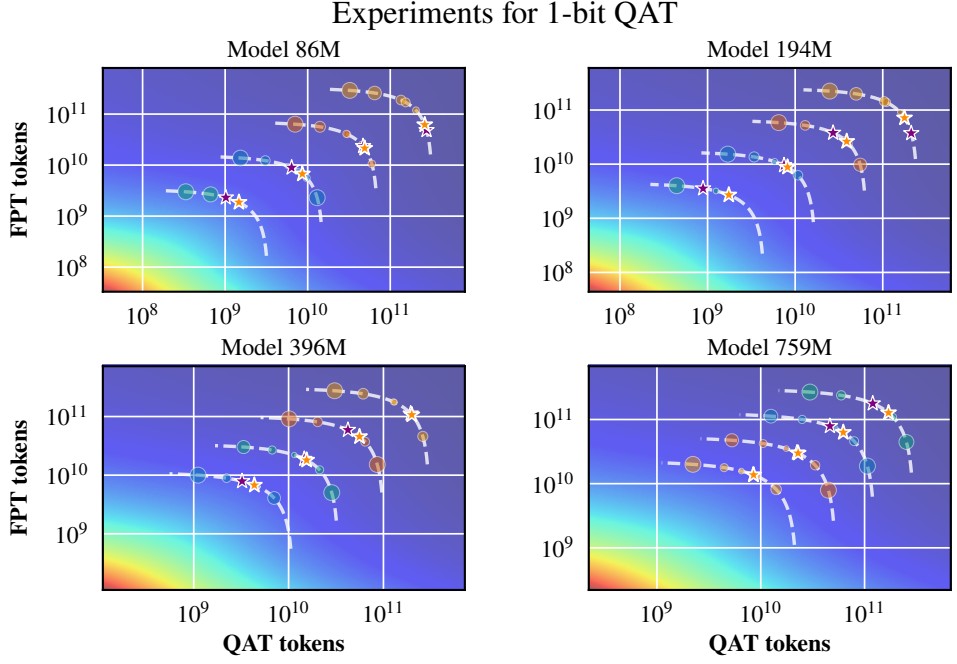

(a) The loss fit metrics are: $R^2$ = 0.982, MAE = 0.026, MAPE = 0.895%. Inferred from loss QAT optimum fraction prediction metrics: MAE = 0.081.

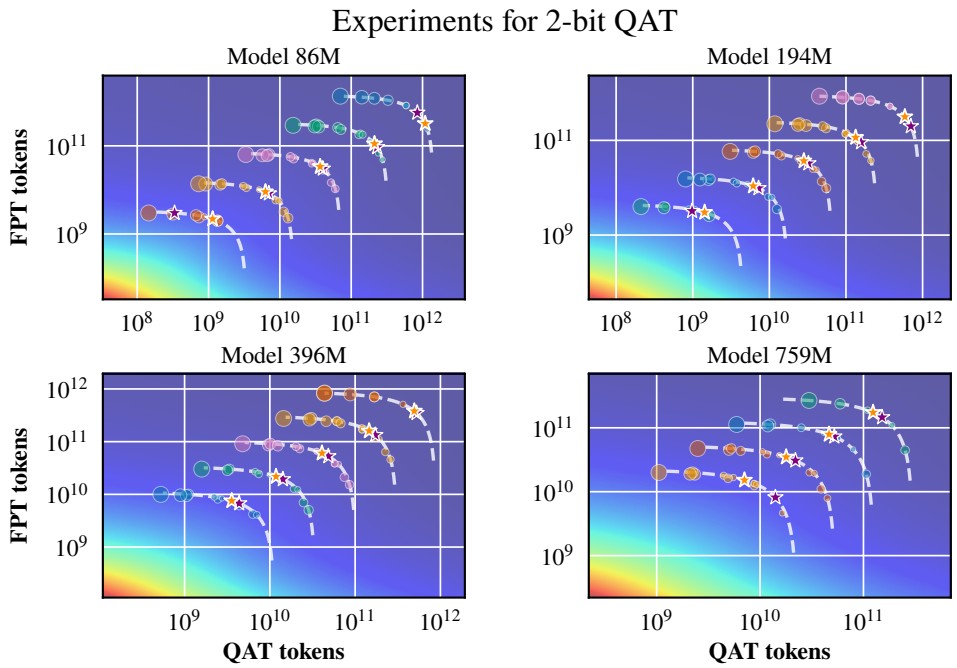

(b) The loss fit metrics are: $R^2$ = 0.981, MAE = 0.023, MAPE = 0.817%. Inferred from loss QAT optimum fraction prediction metrics: MAE = 0.102.

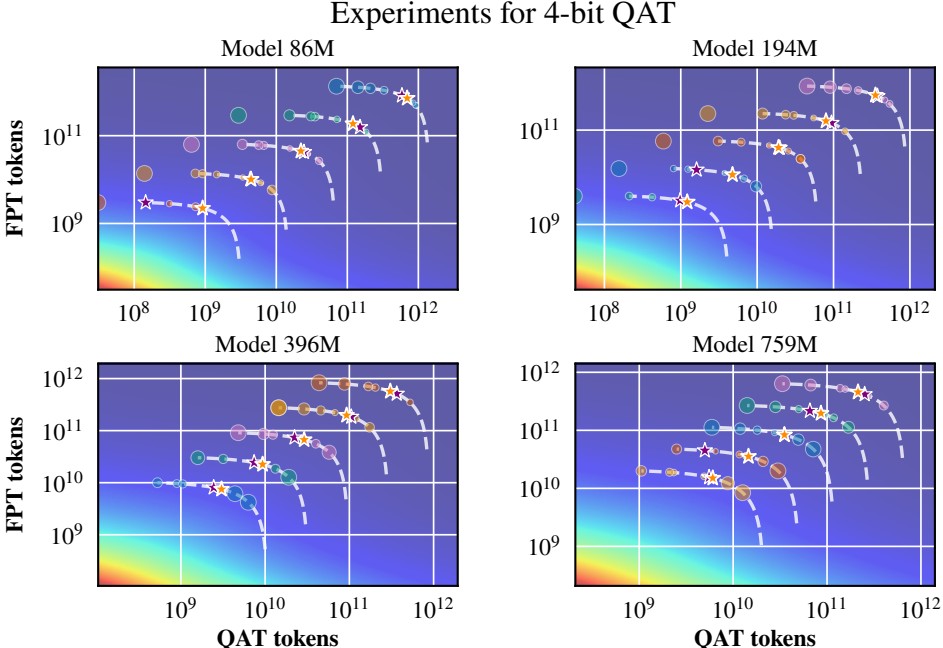

(c) The loss fit metrics are: $R^2 = 0.983$, MAE = 0.021, MAPE = 0.796%. Inferred from loss QAT optimum fraction prediction metrics: MAE = 0.074.

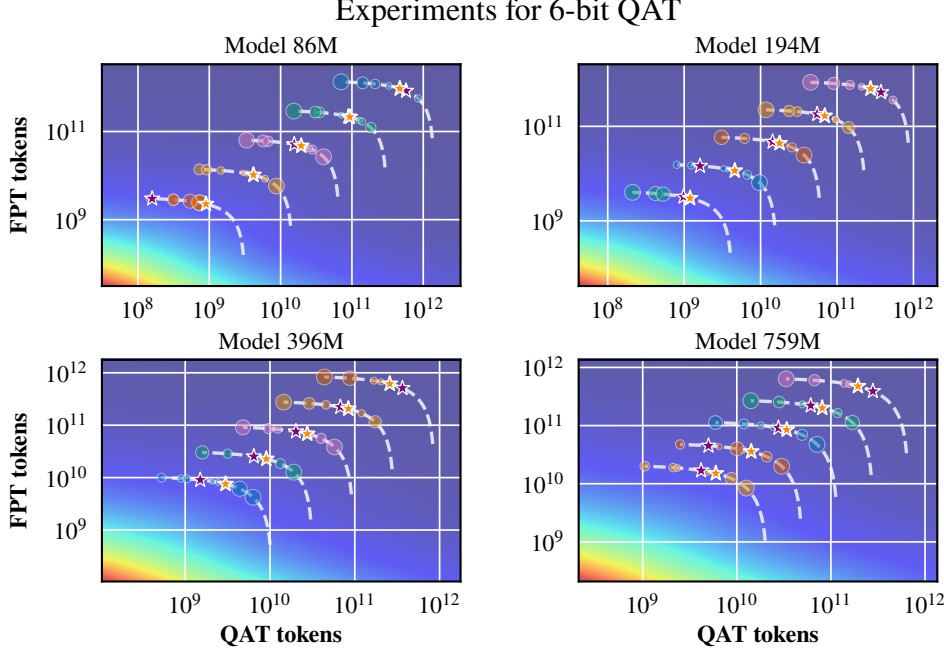

(d) The loss fit metrics are: $R^2 = 0.991$, MAE = 0.018, MAPE = 0.661%. Inferred from loss QAT optimum fraction prediction metrics: MAE = 0.09.

Figure 10: Visualizations of the fitted loss scaling laws for different QAT bit-widths. Experimental data are plotted with point sizes corresponding to loss relative to the group of experiments with the same $D_{\text{total}}$. Orange stars correspond to theoretical optima; purple stars represent experimental optima.

# E    Scaling Law Performance for Low Token Counts

One may notice that the scaling law optimal fraction prediction error is high for low token counts (appendix D). Specifically, the optimal QAT fraction appears to be lower than the predicted one. In this section, we attempt to provide an explanation for this behavior. As low-token setups are not practically important, we do not include this discussion in the main text.

Intuitively, with low $S_{fp}$ the model is severely under-trained, and noise introduced by quantization does not significantly alter learned features. In the extreme case of $S_{fp} \approx 0$, simple QAT initialization is already able to almost completely restore performance. We were able to capture such a drop in optimal fraction for low $S_{fp}$ using a more sophisticated form of the scaling law, but, as we noted previously, this has low practical value. Therefore, we prioritized a simpler scaling law form.

# F    Scaling Law Fit Notes

In this section, we summarize methods implemented to achieve better loss scaling law fits. As noted in the main text, we use Huber loss (Huber, 1964) and gradient descent optimization. The Huber loss choice is consistent with the setup of Hoffmann et al. (2022); Chen et al. (2025b). Additionally, we verified that simple MSE achieves worse generalization. We attribute this phenomenon to the presence of outliers in our experiments—this can be seen from the appendix D figures. Specifically, one can notice both outliers for optimal experimental QAT fraction and disproportionate dot sizes. Also, to facilitate generalization over different bit-widths, we re-weight each sample loss contribution proportionally to the corresponding $B$ inverse frequency.

Another important trick is the addition of full-precision loss regularization. This is done based on the expectation that for high $B$, the final loss should be indistinguishable from the full-precision model loss. Therefore, we add **374** full-precision model evaluation results to the fit, assigning $B = 16$ to them, which brings the total fit data size to **1131** experiments. For $D_{fp}, D_{qat}$ assignment, we notice that only the FP/QAT interaction term of $\delta(N, D_{qat}, D_{fp}, B)$ (equation 2) makes a noticeable contribution with high $B$. Therefore, we assign such $D_{fp}, D_{qat}$ : $D_{fp} + D_{qat} = D_{total}$ that minimize the FP/QAT interaction term only. This way, **the obtained QAT loss scaling law fit not only predicts QAT loss, but also predicts full-precision loss** by using $B = 16$. The fit achieves $R^2 = 0.989$, $MAPE = 0.8\%$, $MAE = 0.022$ fit metrics for all obtained full-precision checkpoints.

# G    Fitted Loss Scaling Law Formulas (Specific Bit-Width)

In figures 11a, 11b, 11c and 11d, we present loss scaling laws fitted to our experiments for each specific bit-width separately and the corresponding fit accuracies. For simplicity, we substitute: $S_{qat} = \frac{D_{qat}}{N \cdot \frac{B}{8}}$, $S_{fp} = \frac{D_{fp}}{N \cdot \frac{B}{8}}$. Additionally, table 5 showcases fit metrics of the unified scaling law (section 3.2) and per-bit-width scaling laws (this section). The fit quality is overall comparable, with fits for each bit-width being slightly better. However, we prioritize the unified scaling law due to its higher practical utility and as a way to reduce fit variance.

$$1.931 + \frac{2605.0}{D_{total}^{0.7155}} + \frac{233.6}{N^{0.2921}} + \frac{366.8}{N^{0.367} \cdot S_{qat}^{0.187}} + \frac{970.4}{N^{0.2338} \cdot S_{fp}^{0.5702} \cdot S_{qat}^{0.2388}}$$

(a) Fitted loss scaling law for **1 bits** QAT bit-width. The loss fit metrics are: $R^2 = 0.99$, MAE = 0.02, MAPE = 0.676%. Inferred from loss QAT optimum fraction prediction metrics: MAE = 0.06.

$$1.885 + \frac{2321.0}{D_{total}^{0.4258}} + \frac{368.2}{N^{0.3434}} + \frac{33.01}{N^{0.2426} \cdot S_{qat}^{0.0269}} + \frac{115.9}{N^{0.1763} \cdot S_{fp}^{0.455} \cdot S_{qat}^{0.2636}}$$

(b) Fitted loss scaling law for **2 bits** QAT bit-width. The loss fit metrics are: $R^2 = 0.989$, MAE = 0.019, MAPE = 0.695%. Inferred from loss QAT optimum fraction prediction metrics: MAE = 0.061.

$$1.923 + \frac{2388.0}{D_{\text{total}}^{0.3917}} + \frac{401.3}{N^{0.3389}} + \frac{983.4}{N^{0.6453} \cdot S_{\text{qat}}^{0.1001}} + \frac{54.46}{N^{0.1323} \cdot S_{\text{fp}}^{0.7778} \cdot S_{\text{qat}}^{0.2755}}$$

(c) Fitted loss scaling law for **4 bits** QAT bit-width. The loss fit metrics are: $R^2 = 0.982$, MAE = 0.02, MAPE = 0.735%. Inferred from loss QAT optimum fraction prediction metrics: MAE = 0.075.

$$1.829 + \frac{1546.0}{D_{\text{total}}^{0.3826}} + \frac{301.4}{N^{0.444}} + \frac{148.5}{N^{0.2853} \cdot S_{\text{qat}}^{0.0004}} + \frac{28.33}{N^{0.1381} \cdot S_{\text{fp}}^{0.5881} \cdot S_{\text{qat}}^{0.1595}}$$

(d) Fitted loss scaling law for **6 bits** QAT bit-width. The loss fit metrics are: $R^2 = 0.992$, MAE = 0.017, MAPE = 0.604%. Inferred from loss QAT optimum fraction prediction metrics: MAE = 0.049.

Figure 11: Fitted loss scaling law formulas, fitted for each QAT bit-width separately.

Table 5: Comparison between unified QAT loss scaling law (section 3.2) and separate loss scaling laws for each bit-width. The fit quality is overall similar, with separate scaling laws achieving slightly better fits.

| $B$ | MAE, loss fit | | $R^2$, loss fit | | MAE, optimal QAT fraction fit | |
|---|---|---|---|---|---|---|
| | **Unified** | **Separate** | **Unified** | **Separate** | **Unified** | **Separate** |
| 1 | 0.026 | 0.02 | 0.982 | 0.99 | 0.081 | 0.06 |
| 2 | 0.023 | 0.019 | 0.981 | 0.989 | 0.102 | 0.061 |
| 4 | 0.021 | 0.02 | 0.983 | 0.982 | 0.074 | 0.075 |
| 6 | 0.018 | 0.017 | 0.991 | 0.992 | 0.09 | 0.049 |

## H  QAT AND FP LOSS SCALING LAWS INTERPLAY

As discussed in appendix F, we fit the QAT scaling law such that $B = 16$ substitution approximates full-precision model loss, so we use this setup to estimate full-precision model accuracy in the section 4.2 analysis.

Points of interest in figure 5 are where lines cross $y = 0$. Such a point represents the maximum $D_{\text{total}}$ for which the corresponding QAT can reproduce FP loss. In tables 6 and 7, we show such values for models from figure 5. We consider a 0.5% QAT/FP perplexity difference to be minor and calculate zero-crossing accounting for this margin. As expected, larger models can maintain FP quality for lower bit-widths and higher total token counts.

Table 6: Token count for figure 5 (**Left**) lines' zero-crossing. This represents the maximum total token count for the **500.0M** model when QAT of the corresponding bit-width can restore FP model quality. "N/A" means that for any token count, the bit-width cannot achieve accuracy similar to the full-precision model.

Table 7: Token count for figure 5 (**Right**) lines' zero-crossing. This represents the maximum total token count for the **16.0B** model when QAT of the corresponding bit-width can restore FP model quality.

| B | Max FP restore tokens count |
|---|---|
| 1 | N/A |
| 2 | N/A |
| 3 | N/A |
| 4 | 83.6B |
| 5 | 1.1T |
| 6 | > 100 T |

| B | Max FP restore tokens count |
|---|---|
| 1 | 80.3B |
| 2 | 212.1B |
| 3 | 633.2B |
| 4 | 2.8T |
| 5 | > 100 T |
| 6 | > 100 T |

# I  OPTIMAL QAT BIT-WIDTH VERIFICATION

Section 4.3 analyzes which $B$ is optimal within specific memory and training compute budgets. We verify the presented plot in figure 12. To do so, we linearly interpolate information from conducted experiments. While such interpolation yields some artifacts, the general structure is consistent with the predicted one. Additionally, we plot loss levels of the optimal QAT selection in figure 13. Results reveal that loss levels closely follow the predicted ones.

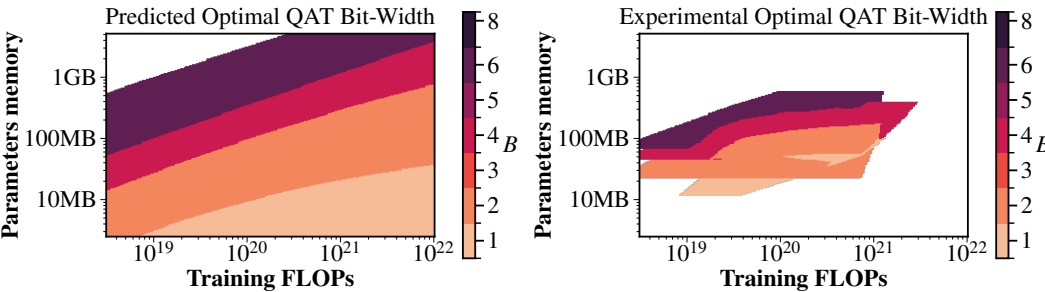

Figure 12: Comparison of predicted optimal QAT bit-width and experimental optima. **On the left**, we reproduce figure 12 but with a reduced set of bit-widths corresponding to the set of bit-widths used in the conducted experiments (1, 2, 4, 6). **On the right**, we show optimal QAT bit-widths obtained from real experimental data. We take experiments with optimal QAT fraction and interpolate the grid into them. The white area represents the range of values where we do not have experimental data. It is clearly seen that the general structure of predicted optima corresponds to the real experimental one.

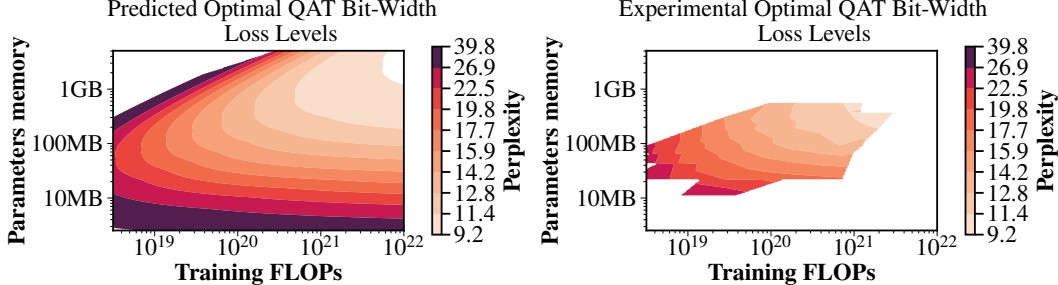

Figure 13: Comparison of predicted optimal QAT bit-width loss levels and experimental ones. The presented figures show loss levels of corresponding optimal QAT configurations from figure 12. We use the same color mapping and normalization for both plots. **On the left**, we show loss levels of figure 12 **(Left)**. **On the right**, we show optimal QAT configuration loss levels obtained from real experimental data. The white area represents the range of values where we do not have experimental data. It is clearly seen that predicted loss levels closely follow the true optimal loss levels. Note that the experimental plot incorporates experiments of different bit-widths as displayed in figure 12 **(Right)**.

## J    DATASET AND HYPERPARAMETER IMPACT

To ensure that the observed phenomenon is not dataset- or hyperparameter-induced, we conduct small-scale 4-bit QAT experiments, pretraining the model on the SlimPajama (Soboleva et al., 2023) dataset with different pretraining batch sizes and learning rate selections (table 8). The results are presented in figure 14; we plot the DCLM-based best fraction prediction fit that was used in the main text. It is clearly seen that the same optimal fraction growth phenomenon is observed, and except for several outliers, the fit is quite accurate. Even with dataset and hyperparameter substitution and no additional fitting, the optimal fraction fit achieves 0.111 MAE. This shows that the conclusions made in the main text are minimally influenced by the exact hyperparameters and dataset choice we made. However, we expect the loss scaling law fit to differ more due to the dependence on data quality as reported by Bi et al. (2024). The optimal QAT fraction inferred from the loss scaling law error is 0.129 MAE.

Table 8: Hyperparameters used during the SlimPajama-based experiment reproduction. We purposefully changed hyperparameters to test how robust the observed phenomenon is.

| Model size, M | Pretrain | | QAT | |
| --- | --- | --- | --- | --- |
| | Batch size | Learning rate | Batch size | Learning rate |
| 86 | 417,792 | 2.0e-04 | 208,896 | 1.0e-04 |
| 194 | 483,328 | 2.0e-04 | 245,760 | 1.0e-04 |
| 396 | 573,440 | 2.0e-04 | 204,800 | 1.0e-04 |
| 759 | 655,360 | 2.0e-04 | 262,144 | 1.0e-04 |

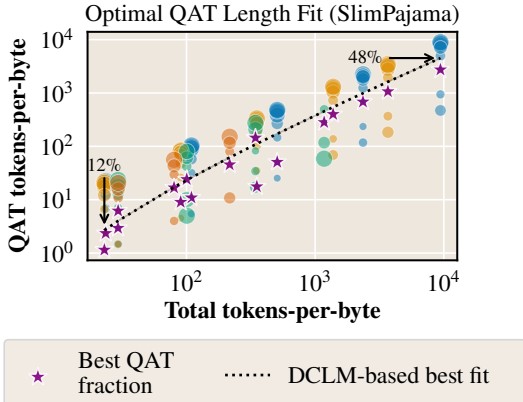

Figure 14: Optimal QAT fraction for SlimPajama-based experiment reproduction. It is clearly seen that the optimal fractions also increase with the total tokens-per-parameter-byte statistic. The fit from the main text (DCLM-based fit) is also plotted for reference. Even without additional re-fitting, the optimal fraction fit achieves 0.111 MAE. This indicates that the observed phenomenon is not dataset- or hyperparameter-induced.

## K  2.2B MODEL OPTIMAL QAT FRACTION PREDICTION

In this section, we verify the scalability of the obtained results. To do so, we train a 2.2B model with QAT using several different QAT fractions, including the predicted optimal QAT fraction. We verify that the predicted optimal QAT fraction from the loss scaling law generalizes to the 2.2B model, which is 2.9 times larger than the largest model in the loss scaling law fit data. The results are presented in table 9.

Table 9: Experiments for the 2.2B parameter model. We select the middle fraction to be close to the predicted optimal one and two additional fractions: one smaller than optimal and one larger. We present the corresponding perplexities and the difference between the minimum perplexity and the perplexity corresponding to the predicted optimal QAT fraction ($L_*$). It is seen that in most cases the predicted QAT fraction is optimal, and in some cases it deviates from the optimum insignificantly— we expect this to be noise.

| $B$ | $D_{\text{total}}$ | Tested Fractions | Perplexities | $\frac{|L_{\min}-L_*|}{L_{\min}}$, % |
|---|---|---|---|---|
| 1 | 49.3B | 10.0%, 38.3%, 53.3% | 13.502, 13.017, 13.092 | 0.00% |
|   | 109.5B | 10.0%, 40.9%, 55.9% | 12.563, 12.187, 12.25 | 0.00% |
| 2 | 22.2B | 10.0%, 39.2%, 54.2% | 13.95, 13.828, 13.734 | 0.68% |
|   | 49.3B | 10.0%, 40.3%, 55.3% | 12.335, 12.068, 12.084 | 0.00% |
| 4 | 22.2B | 10.0%, 26.5%, 41.5% | 13.017, 13.049, 13.198 | 0.24% |
|   | 49.3B | 10.0%, 26.7%, 41.7% | 11.515, 11.515, 11.545 | 0.00% |
| 6 | 20.6B | 2.9%, 17.9%, 32.9% | 13.149, 13.114, 13.21 | 0.00% |

## L    QAT OVERHEAD

In this section, we show results of our benchmarks that measure the slowdown between QAT and FP training. In our benchmarks, we select the maximum batch size that fits within GPU memory constraints and perform multiple measurements to reduce the variance of our results. We do not observe significant slowdown for all model sizes we have tested. Figure 15 summarizes our findings. It is important to note that ensuring that PyTorch (Paszke et al., 2019) compile optimization processed quantization operators correctly and without slow fallbacks was crucial to achieving almost zero overhead.

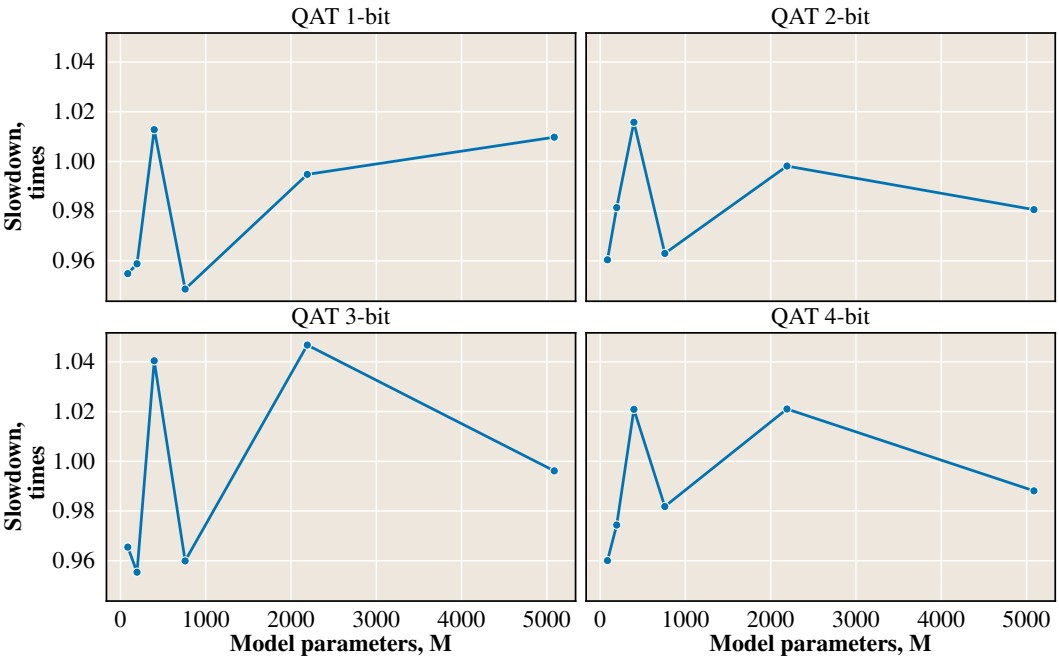

Figure 15: Measured overhead of QAT versus FP training. It is clearly seen that the slowdown fraction fluctuates around 1.0 and no significant slowdown is noticeable.

# M  QAT & LEARNING RATE COOLDOWN FUSION: EXTENDED RESULTS

In this section, we show results of "QAT & Learning Rate Cooldown Fusion" for all bit widths (table 10). As discussed in section 5, the proposed approach shows consistent improvements for 4- and 6-bit QAT. For 1- and 2-bit experiments, improvements in some settings are present but less prominent than for 4- and 6-bit QAT. We explain this by the large optimal QAT fraction for lower bits, which minimizes the impact of QAT & Cooldown Fusion.

Table 10: Accuracy comparison between the classic QAT scheme and "QAT & Learning Rate Cooldown Fusion" scheme. The loss difference is reported in "wasted tokens"—the difference in total token count between optimal QAT fraction loss points in the loss scaling law. Substantial improvements are noticeable across different model sizes and token counts for 4-bit and higher. For 1- and 2-bit experiments, improvements in some settings are present but less prominent. We explain this by the large optimal QAT fraction for lower bits, which minimizes the impact of QAT & Cooldown Fusion.

| $B$ | Model size, M | $D_{\text{total}}$ | Perplexity | | Wasted tokens, ↑ |
| --- | --- | --- | --- | --- | --- |
| | | | Unfused (baseline) | Fused (ours) | Unfused total tokens, % |
| 1 | 74 | 70.4B | **23.82** | $24.14_{+1.34\%}$ | -19.5% |
| | 163 | 17.0B | **20.95** | $21.06_{+0.53\%}$ | -5.3% |
| | 425 | 11.1B | 18.53 | $\textbf{18.43}_{-0.54\%}$ | 4.4% |
| | | 33.5B | **16.33** | $16.41_{+0.49\%}$ | -8.1% |
| | | 305.8B | **14.73** | $14.83_{+0.68\%}$ | -26.8% |
| | 816 | 22.2B | 16.3 | $\textbf{16.17}_{-0.80\%}$ | 7.3% |
| | | 52.8B | **14.83** | $14.9_{+0.47\%}$ | -6.9% |
| 2 | 74 | 15.3B | 21.36 | $\textbf{21.32}_{-0.19\%}$ | 2.1% |
| | | 70.4B | **19.54** | $19.62_{+0.41\%}$ | -7.1% |
| | | 323.6B | **18.66** | $18.74_{+0.43\%}$ | -12.4% |
| | 163 | 17.0B | 18.28 | $\textbf{18.17}_{-0.60\%}$ | 5.6% |
| | | 65.0B | **16.36** | $16.39_{+0.18\%}$ | -2.3% |
| | 425 | 11.1B | 17.01 | $\textbf{16.69}_{-1.88\%}$ | 13.3% |
| | | 33.5B | 14.59 | $\textbf{14.51}_{-0.55\%}$ | 7.8% |
| | | 101.3B | 13.38 | $\textbf{13.37}_{-0.07\%}$ | 2.1% |
| | | 305.8B | **12.65** | $12.66_{+0.08\%}$ | -5.9% |
| | 816 | 52.8B | 13.35 | $\textbf{13.27}_{-0.60\%}$ | 6.5% |
| | | 297.5B | **11.77** | $11.77_{-0.00\%}$ | 2.0% |
| 4 | 74 | 1.4T | 16.26 | $\textbf{16.25}_{-0.06\%}$ | 2.2% |
| | 163 | 901.3B | 13.51 | $\textbf{13.49}_{-0.15\%}$ | 9.2% |
| | 425 | 10.5B | 16.3 | $\textbf{16.02}_{-1.72\%}$ | 9.6% |
| | | 31.8B | 13.9 | $\textbf{13.76}_{-1.01\%}$ | 10.4% |
| | | 96.0B | 12.62 | $\textbf{12.54}_{-0.63\%}$ | 13.6% |
| | 816 | 281.9B | 11.07 | $\textbf{11.02}_{-0.45\%}$ | 13.2% |
| 6 | 74 | 306.6B | 16.45 | $\textbf{16.41}_{-0.24\%}$ | 9.1% |
| | | 1.4T | 15.85 | $\textbf{15.82}_{-0.19\%}$ | 14.3% |
| | 163 | 61.6B | 14.92 | $\textbf{14.83}_{-0.60\%}$ | 9.5% |
| | | 901.3B | 13.21 | $\textbf{13.18}_{-0.23\%}$ | 27.9% |
| | 425 | 31.8B | 13.72 | $\textbf{13.59}_{-0.95\%}$ | 10.4% |
| | | 96.0B | 12.44 | $\textbf{12.36}_{-0.64\%}$ | 15.5% |
| | | 289.7B | 11.63 | $\textbf{11.58}_{-0.43\%}$ | 38.8% |
| | 816 | 118.7B | 11.59 | $\textbf{11.51}_{-0.69\%}$ | 11.4% |
| | | 281.9B | 10.92 | $\textbf{10.85}_{-0.64\%}$ | 16.6% |

# N  EXPERIMENT TOKEN COUNTS

Table 11 summarizes the total token counts used throughout the experiments. For each token count, several $D_{\text{fp}}$ / $D_{\text{qat}}$ ratios were tested. Selected ratios for different setups are displayed in tables 12, 13, 14 and 15. In addition to the reported structured experiments, we conducted experiments with extreme QAT fractions (close to 1% and close to 100%) to improve loss scaling law fitting across the range of different values.

Table 11: List of total token counts analyzed for different model sizes.

| Model Size (M) | Total Tokens |
|---|---|
| 86 | 2.3B, 2.4B, 2.6B, 3.0B, 3.1B, 3.3B, 5.9B, 10.5B, 13.2B, 13.9B, 14.5B, 14.8B, 15.3B, 27.0B, 41.8B, 60.6B, 61.7B, 64.0B, 66.7B, 70.4B, 123.9B, 171.3B, 274.4B, 278.7B, 294.2B, 306.6B, 323.6B, 569.3B, 1.2T, 1.3T, 1.4T |
| 194 | 3.2B, 3.3B, 4.0B, 4.2B, 4.4B, 6.5B, 9.5B, 14.6B, 15.5B, 16.1B, 17.0B, 24.9B, 36.3B, 56.0B, 59.1B, 61.6B, 65.0B, 95.2B, 138.8B, 182.6B, 214.2B, 226.1B, 235.6B, 248.7B, 364.2B, 530.8B, 698.3B, 819.4B, 901.3B |
| 396 | 4.3B, 8.2B, 9.6B, 9.7B, 10.5B, 11.1B, 12.8B, 24.6B, 28.9B, 30.5B, 31.8B, 33.5B, 56.5B, 84.4B, 87.2B, 92.1B, 96.0B, 101.3B, 170.6B, 263.4B, 289.7B, 305.8B, 515.2B, 874.8B |
| 759 | 8.5B, 21.1B, 22.2B, 48.0B, 50.0B, 52.8B, 113.9B, 118.7B, 125.3B, 281.8B, 297.5B, 536.7B, 669.2B |

Table 12: List of different QAT fractions analyzed for the 86M parameter model and different total token counts.

| Model Size (M) | $B$ | $D_{\text{total}}$ | $D_{\text{qat}}/D_{\text{total}}$ |
|---|---|---|---|
| 86 | 1 | 3.3B | 10.0%, 20.0%, 30.6% |
| | 1 | 15.3B | 10.0%, 20.0%, 37.3%, 41.7%, 49.7%, 63.3%, 85.0% |
| | 1 | 70.4B | 10.0%, 20.0%, 41.7%, 42.9%, 63.3%, 66.6%, 85.0% |
| | 1 | 323.6B | 10.0%, 20.0%, 41.7%, 47.6%, 63.3%, 85.0% |
| | 2 | 3.1B | 4.6%, 10.0%, 20.0%, 23.3%, 26.3%, 40.0% |
| | 2 | 14.5B | 5.0%, 6.3%, 10.0%, 20.0%, 23.3%, 33.5%, 41.7%, 45.7%, 60.0%, 63.3%, 78.0%, 85.0% |
| | 2 | 66.7B | 5.0%, 8.5%, 10.0%, 20.0%, 23.3%, 39.6%, 41.7%, 56.9%, 60.0%, 63.3%, 78.0%, 85.0% |
| | 2 | 306.6B | 5.0%, 10.0%, 11.4%, 20.0%, 23.3%, 41.7%, 44.7%, 60.0%, 63.3%, 71.3%, 78.0%, 85.0% |
| | 2 | 1.4T | 5.0%, 10.0%, 15.0%, 23.3%, 41.7%, 60.0%, 78.0%, 83.2% |
| | 4 | 3.0B | 1.0%, 4.6%, 10.0%, 23.3%, 26.7% |
| | 4 | 13.9B | 1.0%, 5.0%, 6.3%, 10.0%, 23.3%, 29.7%, 41.7%, 60.0% |
| | 4 | 64.0B | 1.0%, 5.0%, 8.5%, 10.0%, 23.3%, 36.8%, 41.7%, 60.0% |
| | 4 | 123.9B | 1.0%, 50.0%, 90.8% |
| | 4 | 274.4B | 1.0%, 50.0%, 79.7% |
| | 4 | 294.2B | 1.0%, 5.0%, 10.0%, 11.4%, 23.3%, 41.7%, 49.2%, 60.0% |
| | 4 | 569.3B | 1.0%, 50.0%, 57.9% |
| | 4 | 1.2T | 1.0%, 10.5% |
| | 4 | 1.4T | 5.0%, 10.0%, 15.0%, 23.3%, 41.7%, 60.0%, 65.0% |
| | 6 | 3.1B | 4.6%, 5.0%, 10.0%, 17.0%, 20.0%, 23.3%, 23.4% |
| | 6 | 14.5B | 5.0%, 6.3%, 10.0%, 20.8%, 23.3%, 28.6%, 41.7%, 60.0% |
| | 6 | 66.7B | 5.0%, 8.5%, 10.0%, 23.0%, 23.3%, 38.0%, 41.7%, 60.0% |
| | 6 | 171.3B | 1.0%, 50.0%, 87.3% |
| | 6 | 306.6B | 5.0%, 10.0%, 11.4%, 23.3%, 29.0%, 41.7%, 45.3%, 60.0% |
| | 6 | 1.4T | 5.0%, 10.0%, 15.0%, 23.3%, 40.7%, 41.7%, 60.0% |

Table 13: List of different QAT fractions analyzed for the 194M parameter model and different total token counts.

| Model Size (M) | $B$ | $D_{\text{total}}$ | $D_{\text{qat}}/D_{\text{total}}$ |
|---|---|---|---|
| 194 | 1 | 4.4B | 10.0%, 20.0%, 28.1% |
| | 1 | 17.0B | 10.0%, 20.0%, 34.3%, 41.7%, 43.9%, 63.3% |
| | 1 | 65.0B | 10.0%, 20.0%, 39.7%, 41.7%, 56.4%, 63.3%, 85.0% |
| | 1 | 248.7B | 10.0%, 20.0%, 41.7%, 44.4%, 63.3%, 85.0% |
| | 2 | 4.2B | 5.0%, 10.0%, 20.0%, 23.3%, 23.6%, 38.8% |
| | 2 | 16.1B | 5.0%, 7.7%, 10.0%, 20.0%, 23.3%, 30.3%, 41.7%, 42.5%, 60.0%, 63.3%, 78.0% |
| | 2 | 61.6B | 5.0%, 10.0%, 20.0%, 23.3%, 36.1%, 41.7%, 49.7%, 60.0%, 63.3%, 78.0%, 85.0% |
| | 2 | 235.6B | 5.0%, 10.0%, 12.8%, 20.0%, 23.3%, 41.1%, 41.7%, 60.0%, 61.0%, 63.3%, 78.0%, 85.0% |
| | 2 | 901.3B | 5.0%, 10.0%, 16.4%, 23.3%, 41.7%, 78.0% |
| | 4 | 4.0B | 1.0%, 5.0%, 10.0%, 23.3% |
| | 4 | 15.5B | 1.0%, 5.0%, 7.7%, 10.0%, 23.3%, 27.9%, 41.7%, 60.0% |
| | 4 | 56.0B | 1.0%, 50.0%, 93.5% |
| | 4 | 59.1B | 1.0%, 5.0%, 10.0%, 23.3%, 32.0%, 41.7%, 60.0% |
| | 4 | 95.2B | 1.0%, 50.0%, 89.0% |
| | 4 | 138.8B | 1.0%, 84.0% |
| | 4 | 182.6B | 1.0%, 50.0%, 78.9% |
| | 4 | 214.2B | 1.0%, 50.0%, 75.2% |
| | 4 | 226.1B | 1.0%, 5.0%, 10.0%, 12.8%, 23.3%, 39.9%, 41.7%, 60.0% |
| | 4 | 364.2B | 1.0%, 50.0%, 57.9% |
| | 4 | 530.8B | 1.0%, 38.6% |
| | 4 | 901.3B | 5.0%, 10.0%, 16.4%, 23.3%, 41.7%, 51.9%, 60.0% |
| | 6 | 4.2B | 5.0%, 10.0%, 12.6%, 23.3% |
| | 6 | 16.1B | 5.0%, 7.7%, 10.0%, 20.3%, 23.3%, 41.7%, 60.0% |
| | 6 | 61.6B | 5.0%, 10.0%, 21.6%, 23.3%, 32.8%, 41.7%, 60.0% |
| | 6 | 235.6B | 5.0%, 10.0%, 12.8%, 23.3%, 24.4%, 40.3%, 41.7%, 60.0% |
| | 6 | 901.3B | 5.0%, 10.0%, 16.4%, 23.3%, 41.7%, 60.0% |

Table 14: List of different QAT fractions analyzed for the 396M parameter model and different total token counts.

| Model Size (M) | $B$ | $D_{\text{total}}$ | $D_{\text{qat}}/D_{\text{total}}$ |
|---|---|---|---|
| 396 | 1 | 11.1B | 10.0%, 20.0%, 29.1%, 37.9%, 41.7%, 63.3% |
| | 1 | 33.5B | 10.0%, 20.0%, 34.2%, 41.7%, 43.6%, 63.3%, 85.0% |
| | 1 | 101.3B | 10.0%, 20.0%, 38.7%, 41.7%, 53.5%, 63.3%, 85.0% |
| | 1 | 305.8B | 10.0%, 20.0%, 41.7%, 42.7%, 63.3%, 65.9%, 85.0% |
| | 2 | 10.5B | 5.0%, 8.5%, 10.0%, 20.0%, 23.3%, 24.7%, 39.3%, 41.7%, 60.0%, 63.3% |
| | 2 | 31.8B | 5.0%, 10.0%, 20.0%, 23.3%, 30.1%, 41.7%, 42.3%, 60.0%, 63.3%, 78.0%, 85.0% |
| | 2 | 96.0B | 5.0%, 10.0%, 12.9%, 20.0%, 23.3%, 35.0%, 41.7%, 47.9%, 60.0%, 63.3%, 78.0%, 85.0% |
| | 2 | 289.7B | 5.0%, 10.0%, 15.8%, 20.0%, 23.3%, 39.3%, 41.7%, 56.4%, 60.0%, 63.3%, 78.0%, 85.0% |
| | 2 | 874.8B | 5.0%, 10.0%, 19.2%, 41.7%, 60.0% |
| | 4 | 10.5B | 5.0%, 8.5%, 10.0%, 23.3%, 26.3%, 41.7%, 60.0% |
| | 4 | 31.8B | 5.0%, 10.0%, 23.3%, 27.9%, 41.7%, 60.0% |
| | 4 | 96.0B | 5.0%, 10.0%, 12.9%, 23.3%, 31.0%, 41.7%, 60.0% |
| | 4 | 170.6B | 1.0%, 50.0%, 79.7% |
| | 4 | 263.4B | 1.0%, 50.0%, 68.6% |
| | 4 | 289.7B | 5.0%, 10.0%, 15.8%, 23.3%, 36.3%, 41.7%, 60.0% |
| | 4 | 515.2B | 1.0%, 38.6% |
| | 4 | 874.8B | 5.0%, 10.0%, 19.2%, 23.3%, 41.7%, 60.0% |
| | 6 | 10.5B | 5.0%, 8.5%, 10.0%, 14.3%, 19.8%, 23.3%, 41.7%, 60.0% |
| | 6 | 31.8B | 5.0%, 10.0%, 20.3%, 23.2%, 23.3%, 41.7%, 60.0% |
| | 6 | 96.0B | 5.0%, 10.0%, 12.9%, 21.1%, 23.3%, 31.0%, 41.7%, 60.0% |
| | 6 | 289.7B | 5.0%, 10.0%, 15.8%, 22.9%, 23.3%, 41.7%, 60.0% |
| | 6 | 874.8B | 5.0%, 10.0%, 19.2%, 23.3%, 41.7% |

Table 15: List of different QAT fractions analyzed for the 759M parameter model and different total token counts.

| Model Size (M) | $B$ | $D_{\text{total}}$ | $D_{\text{qat}}/D_{\text{total}}$ |
|---|---|---|---|
| 759 | 1 | 22.2B | 10.0%, 20.0%, 29.3%, 38.0%, 41.7%, 63.3% |
| | 1 | 52.8B | 10.0%, 20.0%, 33.3%, 41.7%, 42.3%, 63.3%, 85.0% |
| | 1 | 125.3B | 10.0%, 20.0%, 37.0%, 41.7%, 63.3%, 85.0% |
| | 1 | 297.5B | 10.0%, 20.0%, 40.3%, 41.7%, 58.1%, 63.3%, 85.0% |
| | 2 | 21.1B | 5.0%, 10.0%, 11.1%, 20.0%, 23.3%, 24.9%, 39.4%, 41.7%, 63.3%, 78.0% |
| | 2 | 50.0B | 5.0%, 10.0%, 13.0%, 20.0%, 23.3%, 29.2%, 41.6%, 41.7%, 60.0%, 63.3%, 78.0%, 85.0% |
| | 2 | 118.7B | 5.0%, 10.0%, 20.0%, 33.2%, 41.7%, 45.4%, 63.3%, 85.0% |
| | 2 | 297.5B | 10.0%, 20.0%, 36.8%, 41.7%, 50.9%, 63.3%, 85.0% |
| | 4 | 21.1B | 5.0%, 10.0%, 11.1%, 23.3%, 26.3%, 41.7%, 60.0% |
| | 4 | 50.0B | 5.0%, 10.0%, 13.0%, 23.3%, 27.6%, 41.7%, 60.0% |
| | 4 | 118.7B | 5.0%, 10.0%, 15.2%, 23.3%, 29.5%, 41.7%, 60.0% |
| | 4 | 281.8B | 5.0%, 10.0%, 17.7%, 23.3%, 32.7%, 41.7%, 60.0% |
| | 4 | 536.7B | 1.0%, 16.4% |
| | 4 | 669.2B | 5.0%, 10.0%, 20.6%, 23.3%, 37.6%, 41.7%, 60.0% |
| | 6 | 21.1B | 5.0%, 10.0%, 11.1%, 19.8%, 23.3%, 41.7%, 60.0% |
| | 6 | 50.0B | 5.0%, 10.0%, 13.0%, 20.2%, 23.3%, 41.7%, 60.0% |
| | 6 | 118.7B | 5.0%, 10.0%, 15.2%, 20.6%, 23.3%, 41.7%, 60.0% |
| | 6 | 281.8B | 5.0%, 10.0%, 17.7%, 21.6%, 23.3%, 41.7%, 60.0% |
| | 6 | 669.2B | 5.0%, 10.0%, 20.6%, 23.3%, 41.7% |

## O  WASTED TOKENS COUNT FORMULATION (SECTION 5)

In this section, we formalize how wasted tokens are calculated for table 2. Let us have loss of fused and unfused experiments for some $D_{\text{total}}$: $L_{\text{fused}}$ and $L_{\text{unfused}}$. Then, similarly to wasted tokens formulation from section 3.2 we can calculate token-distance between $L_{\text{fused}}$ and $L_{\text{unfused}}$ on QAT optimality curve:

$$D^*_{\text{total}}(N, B, l) = \underset{\substack{D'_{\text{total}} \in \mathbb{N} \\ D'_{\text{qat}} = D^*_{\text{qat}}(N, D'_{\text{total}}, B)}}{\arg\min} \left| L(N, D'_{\text{qat}}, D'_{\text{total}} - D'_{\text{qat}}, B) - l \right|,$$

$$D_{\text{wasted}} = D^*_{\text{total}}(N, B, L_{\text{fused}}) - D^*_{\text{total}}(N, B, L_{\text{unfused}}),$$

and the reported percentage is the fraction of unfused total tokens: $\frac{D_{\text{wasted}}}{D^*_{\text{total}}(N, B, L_{\text{unfused}})}$.

## P    QAT ACCURACY

In figures 16a, 16b, 16c and 16d, we plot how optimal QAT fraction experiments compare to the full-precision model with the same total token count. Results reveal that the optimal QAT fraction in 4-bit and 6-bit settings achieves loss close to the full-precision counterpart.

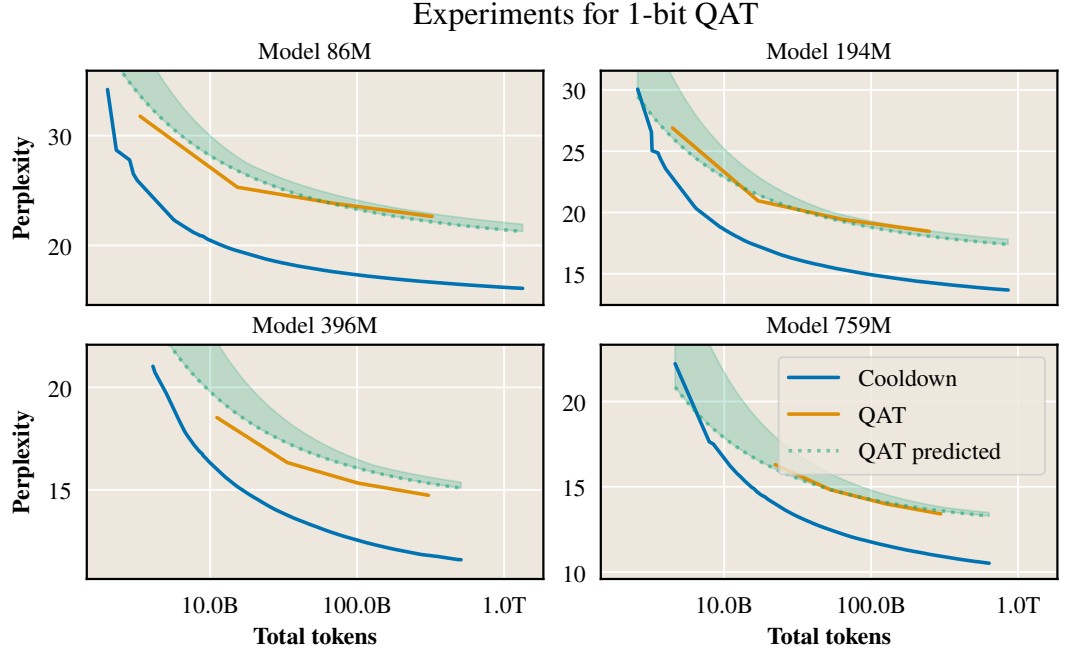

(a) Final loss plots for 1-bit QAT.

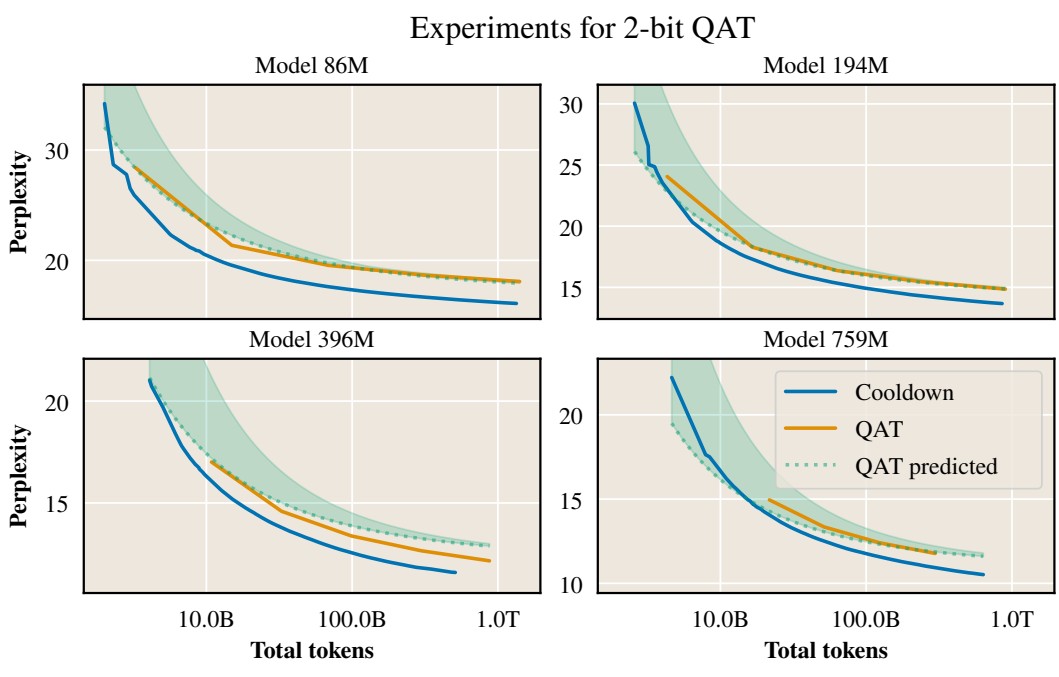

(b) Final loss plots for 2-bit QAT.

## Experiments for 4-bit QAT

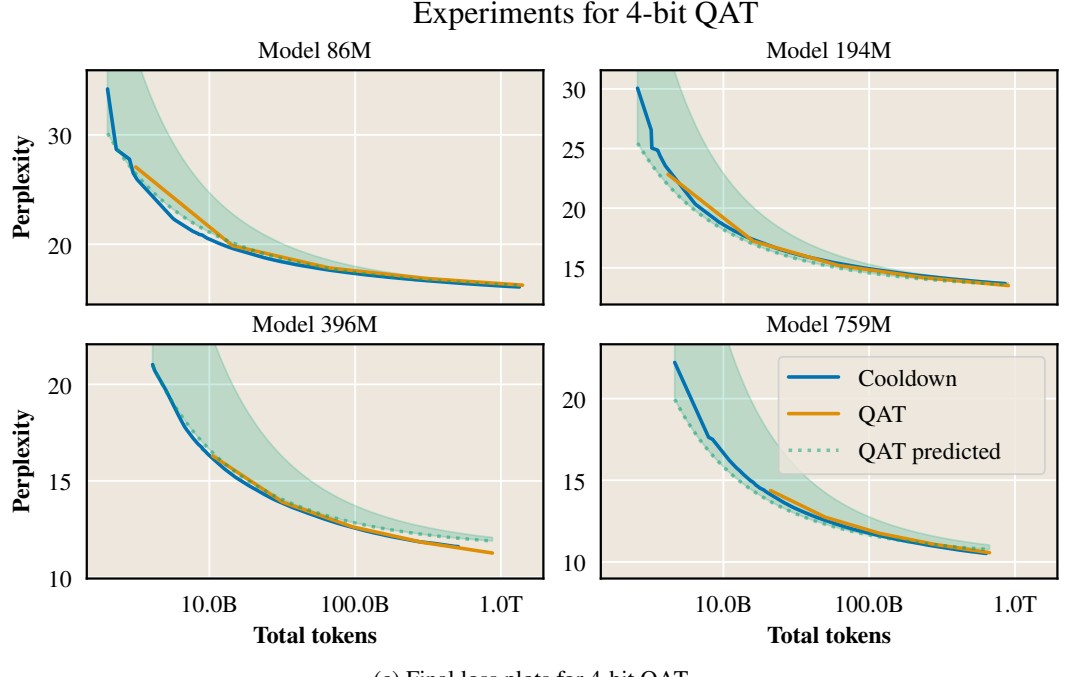

(c) Final loss plots for 4-bit QAT.

## Experiments for 6-bit QAT

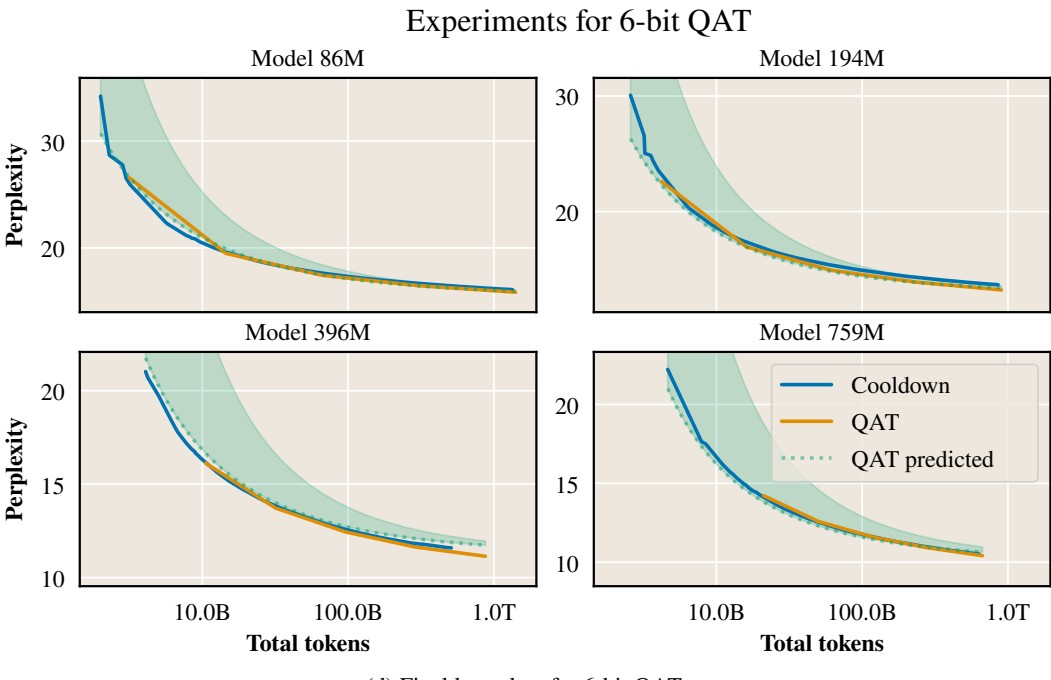

(d) Final loss plots for 6-bit QAT.

Figure 16: Final loss of QAT compared to the full-precision post-cooldown model for the same total token count. For QAT, we plot the best loss for the total token count (optimal QAT fraction experiments). Additionally, we plot the loss predicted for the optimal QAT fraction from the appropriate loss scaling law, and confidence bands correspond to the predicted range of QAT loss for the 5–95% range of QAT fraction.

## Q  UNCERTAINTY ANALYSIS

We analyze fit uncertainty and parameter significance from the perspective of their influence on loss model fit metrics. Formally, we can formulate the problem as follows:

$$H_0 : S(m(\theta = 0)) = S(m(\theta \neq 0)), \quad H_1 : S(m(\theta = 0)) < S(m(\theta \neq 0)),$$

where $S$ is a fit metric of interest for some model, $m(\theta = 0)$ represents a fitted model with parameter $\theta$ forced to zero, and $m(\theta \neq 0)$ represents a fitted model with parameter $\theta$ allowed to vary. We will analyze two metrics: $R^2$ — a general goodness-of-fit metric, and QAT optimal fraction fit MAE (the inequality in $H_1$ is reversed). Together, these two metrics capture two important properties of the scaling law: the ability to predict final model accuracy and the ability to predict the optimal QAT fraction accurately.

To estimate the distribution of the fit metric, we use bootstrapping. We employ the following scheme:

1. Generate a bootstrapped dataset by sampling with replacement from the original dataset.
2. Fit both models $m(\theta = 0), m(\theta \neq 0)$ to the bootstrapped dataset. This step is repeated for several model initialization seeds, and the best fit is selected.
3. Calculate the fit metric for both models.
4. Repeat steps 1–3 $B = 100$ times for each model parameter.

In the end, for each parameter, we obtain two metrics for each bootstrapped dataset corresponding to $m(\theta = 0), m(\theta \neq 0)$. Then, we calculate the difference of metrics and calculate a one-sided 95% quantile confidence interval of the difference. We conclude that the parameter is significant if 0 is not covered by the interval, which means that the model with the parameter is significantly better than the model without it.

Results are presented in figure 17. Combining results for both metrics, all parameters except those corresponding to constant shifts ($\alpha$, $\kappa$, and $\theta$) are significant. This result is expected for QAT fraction MAE, as constant shifts affect the absolute loss value but not the relative position of the optimal QAT fraction (the argmin over a curve). However, this is not expected for $R^2$. Nonetheless, we retain those parameters as they have clear conceptual meaning: $\alpha$ comes from the Chinchilla scaling law, and $\kappa$, $\theta$ model irreducible QAT error. What is more important is that two other added terms in the equation 2 ("pure QAT penalty" and "FP / QAT interaction") are significant.

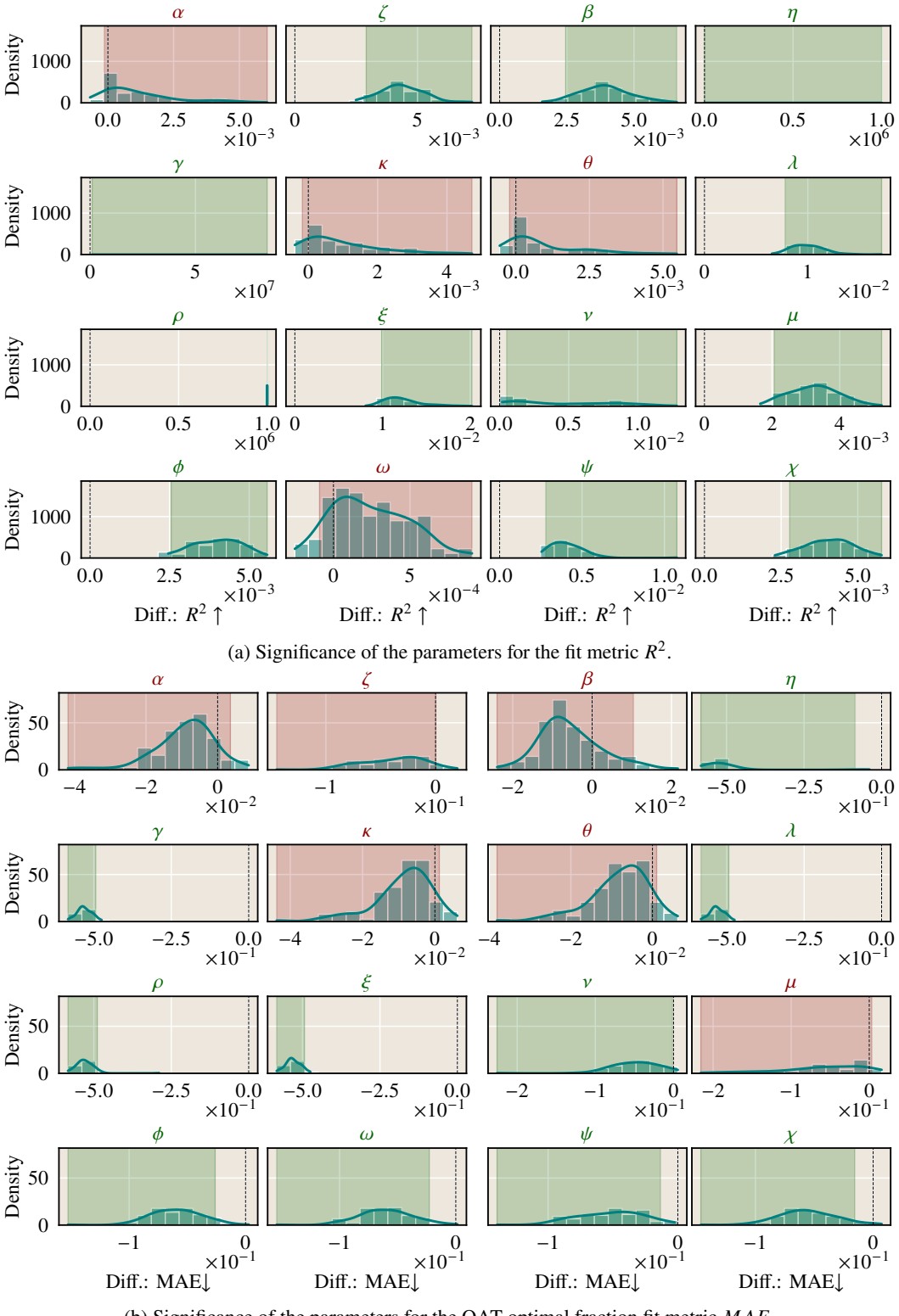

(a) Significance of the parameters for the fit metric $R^2$.

(b) Significance of the parameters for the QAT optimal fraction fit metric $MAE$.

Figure 17: Shaded area represents one-sided 95% quantile confidence interval of the difference in metrics corresponding to constrained and unconstrained models $m(\theta = 0)$ and $m(\theta \neq 0)$ (where $\theta$ is the parameter of interest, indicated by the subplot title). Green color means that 0 is not covered by the interval, meaning that the parameter is significant. Red color means that 0 is covered by the interval, meaning that the parameter's insignificance is not rejected.

# R    FUTURE WORK

In this section, we speculate on possible results for the future work directions proposed in the paper (section 6).

## R.1    PRETRAIN PRECISION & QAT PRECISION INTERACTION

The question of interest is **"How do QAT scaling laws change when pretrain precision is reduced?"** Specifically, a practically important question is how optimal QAT compute allocation changes. Kumar et al. (2025) analyze this question in the context of post-training quantization. While QAT and PTQ yield significant differences in accuracy (especially for lower bits (Liu et al., 2025)), we expect general trends to be similar.

Kumar et al. (2025) report that "overall, models trained in lower precision are more robust to post-training quantization in the sense of incurring lower degradation." We expect the same phenomenon in the context of QAT. Therefore, one may expect the optimal QAT fraction to be smaller when a model is pretrained in lower floating-point precisions (fp4, fp8) than in high precision (fp16, bf16, fp32). Still, we expect the optimal QAT fraction to grow with increasing total compute.

## R.2    QAT SCALING LAW FOR MULTI-STAGE PRETRAINING

Current state-of-the-art chat models commonly incorporate multiple training stages. Commonly, after general cross-entropy pretraining, additional supervised fine-tuning (SFT) and reinforcement learning stages are performed (DeepSeek-AI et al., 2025; OLMo et al., 2025; Apertus et al., 2025; Ouyang et al., 2022; Allal et al., 2025; Rafailov et al., 2023; Schulman et al., 2017; Zhou et al., 2025a). This raises not only the question of how much compute to allocate for QAT but also how to distribute this compute among different stages.

A possible solution is to conduct all post-pretraining stages over the QAT model. Usually, post-training constitutes a minor percentage of compute when compared to pretraining (DeepSeek-AI et al., 2025; Allal et al., 2025; OLMo et al., 2025; Apertus et al., 2025). Therefore, it is natural to expect the optimal QAT fraction to be larger than the entire post-pretraining stage. This means that it is possible to start QAT during pretraining and finish QAT with post-pretraining tuning.

Such a methodology is also motivated by the fact that QAT incurs representation changes, especially in the case of small QAT bit-widths (Liu et al., 2025). Therefore, we believe it is beneficial not to postpone this process of representation change until after post-pretraining stages.

