# OpenReview forum: "Compute-Optimal Quantization-Aware Training"
_ICLR.cc/2026/Conference — ICLR 2026 Poster_

### Official Review · Reviewer_1gxP · 2025-10-28

**Soundness:** 3
**Presentation:** 2
**Contribution:** 3
**Rating:** 8
**Confidence:** 4

**Summary:**

This paper extends scaling-law analysis to the two-stage full-precision & QAT pipeline, asking how to split a fixed training budget between full-precision pretraining and quantisation-aware training. Through hundreds of runs across model sizes (86M–2.2B) and bit-widths, the authors show that the loss-optimal QAT fraction increases with scale, and that this dependency is well captured by a simple _tokens-per-parameter-byte_ statistic. They fit a unified loss scaling law (over ($N$, $D_{fp}$, $D_{qat}$, $B$) that includes pure-QAT and FP/QAT interaction terms, and use it to predict optimal QAT ratios as well as when low-bit QAT can match FP loss. The study also proposes a practical “cooldown & QAT fusion” training schedule (merging LR decay with QAT) and reports small but consistent savings in “wasted tokens” at moderate/high bit-widths. Overall, the paper moves the literature from “QAT should be ~10%” heuristics to compute-aware prescriptions grounded in scaling behaviour, while relying on a strong, modern QAT baseline (ParetoQ)

**Strengths:**

All in all, this is a solid experimental paper that answers fundamental questions in LLM deployment with a sound methodology. I enjoyed how the authors used their loss scaling curves to answer interesting questions in their discussion. For example, the question of what the optimal QAT bitwidth per model and training budget is has been a topic in the community for years. The main strengths are:

* The paper addresses a very significant and unanswered question in LLM inference and of increasing importance for LLM deployment. What budget should be allocated to QAT vs FP training? No previous work has answered this important question in experimentally sound ways, and this work paves the way for significant future work.
* The loss scaling laws are solid, based on hundreds of experiments, and provide strong transparency on the fit of the curve. The introduction of the token-per-byte statistics is also a very useful metric capturing the idiosyncrasies of QAT. The vast corpus of QAT experiments makes for a solid experimental methodology.
* A very extensive appendix, including many results, experimental set-ups, and curves.

**Weaknesses:**

* The paper is somewhat too long given the level of novelty and contribution, and as a result, the central narrative becomes hard to follow. While each section presents interesting findings, the overall flow sometimes feels like a set of independent conclusions about QAT rather than a single, coherent story. I found myself re-reading parts to connect the results to the main scaling-law argument.

* Section 7 reads as an addendum: it explores LR-scheduler fusion, which is interesting but tangential to the main scaling-law contribution, and the reported gains are relatively small and bit-width dependent. I would think this should be part of another paper or appendix.

* The full-precision loss regularisation used to fit the unified law is crucial to interpreting Fig. 5 and the FP-vs-QAT comparisons, but it’s only described in Appendix E. Similarly, the “wasted token count” metric is under-explained and would benefit from a formal definition and more apparent motivation.

**Questions:**

* Figure 1 (right), add $D_{fq}$ and $D_{qat}$ along the axes and include a loss heatmap in the legend
* Equation 2: more discussion and motivation for the forms chosen related to QAT.. What is pure QAT penalty, and why does it take form? * * Why does FP/QAT interaction take this form? Are these grounded in theory or chosen for fitting reasons? Did you do any ablation studies on the form?
* Appendix D. Figure 9: I did not see any discussion or explanation on why the theoretical and experimental optima diverge so much at the lowest $D_{total}$ iso curves for 2,4, and 6 bits for the smallest model of 86M.
* Line 277: wasted token count: I had to read the definition multiple times to understand it. Could you add an equation?
* Line 322: iso-flop. Capitalise flop because it’s hard to understand it when it’s in lower case.
* Figure 5: Make the colour of the bitwidths 5 and 1 more distinct. It’s very hard to tell which curve corresponds to each bitiwidthth
* Section 7: wasted unfused total tokens: could you maybe add an equation? I don’t fully understand how you reach that number.

---

> ### Author Response · Authors · 2025-11-19
> **Addressing weaknesses and questions**
>
> We thank the reviewer for the provided feedback and the high rating of our work.
> In this response we address the weaknesses and questions mentioned in the review.
>
> 1.
> > Figure 1 (right), add $D_\text{qat}$ and $D_\text{fp}$ along the axes and include a loss heatmap in the legend.
>
> > Line 322: iso-flop. Capitalize FLOP because it’s hard to understand when it’s in lowercase.
>
> > Figure 5: Make the colors of the bitwidths 5 and 1 more distinct. It’s very hard to tell which curve corresponds to each bitwidth.
>
> > Section 7: wasted unfused total tokens: could you maybe add an equation? I don’t fully understand how you reach that number.
>
> > Line 277: wasted token count: I had to read the definition multiple times to understand it. Could you add an equation?
>
>
> Thank you for these suggestions. We added the corresponding fixes in the newly uploaded PDF.
>
> 2.
> > Appendix D. Figure 9: I did not see any discussion or explanation on why the theoretical and experimental optima diverge so much at the lowest iso curves for 2, 4, and 6 bits for the smallest model of 86M.
>
>
> That is true; we did not include a possible explanation in the main text as low-token setups are not practically important.
> We provide a plausible explanation in the newly added Appendix E ("Scaling Law Performance for Low Token Counts").
>
> 3.
> > The paper is somewhat too long given the level of novelty and contribution, and as a result, the central narrative becomes hard to follow.
> > While each section presents interesting findings, the overall flow sometimes feels like a set of independent conclusions about QAT rather than a single, coherent story.
> > I found myself re-reading parts to connect the results to the main scaling-law argument.
>
> > Section 7 reads as an addendum: it explores LR-scheduler fusion, which is interesting but tangential to the main scaling-law contribution, and the reported gains are relatively small and bit-width dependent. I would think this should be part of another paper or appendix.
>
>
> We were hoping to organize our findings in the following way:
> - Scaling law fit (Section 4)
> - What can we infer from the scaling law? (Sections 5 and 6)
> - How can we improve QAT efficiency beyond the optimal QAT fraction? (Section 7)
>
> We will be happy to hear your suggestions on how to make our narrative clearer.
>
> 4.
> > The full-precision loss regularization used to fit the unified law is crucial to interpreting Fig. 5 and the FP-vs-QAT comparisons, but it’s only described in Appendix E.
>
> We have added a summary of the method described in the appendix to the main text.
>
> 5.
> > Similarly, the “wasted token count” metric is under-explained and would benefit from a formal definition and more apparent motivation.
>
> > Section 7: wasted unfused total tokens: could you maybe add an equation? I don’t fully understand how you reach that number.
>
> We have added formulations for the "wasted token count" metric.
>
> 6.
> > Equation 2: more discussion and motivation for the forms chosen related to QAT.
> > What is the pure QAT penalty, and why does it take that form?
> > Why does the FP/QAT interaction take this form?
> > Are these grounded in theory or chosen for fitting reasons? Did you do any ablation studies on the form?
>
> We could not build a theoretical framework behind the equation form. As in previous works on scaling laws, we base our form on empirical observations.
>
> Yes, we experimented with many different forms, trying to simplify the equation as much as possible and reduce the number of fitted parameters.
> The current form is the trade-off we found between fit accuracy and equation complexity.
> Specifically, we could not obtain decent results with simpler equations.

---

### Official Review · Reviewer_w7Wt · 2025-10-30

**Soundness:** 3
**Presentation:** 3
**Contribution:** 3
**Rating:** 6
**Confidence:** 3

**Summary:**

This paper investigates the optimal resource allocation ratio between Full-Precision (FP) and Quantization-Aware Training (QAT) phases. Through extensive LLM experiments, the authors found that the optimal QAT fraction increases with the total compute budget and introduced tokens-per-parameter-byte as a key metric to accurately predict this optimum. Loss Scaling Law is proposed to predict the performance of QTA.

**Strengths:**

This paper establishes a novel loss scaling law capable of predicting both the optimal QAT fraction and final model performance across varying bit widths and allocation strategies, demonstrating high practical utility. The proposed method integrates learning rate decay with QAT, effectively eliminating redundant model updates and improving both computational efficiency and accuracy.

**Weaknesses:**

Experiments are conducted on a specific LLM architecture（llama2）, and the scaling law parameters may require refitting for other model types. Different types of models may exhibit varying degrees of tolerance to quantization, particularly those with inherent redundancy.

**Questions:**

How sensitive are the fitted parameters to the range of model sizes and token counts used in the experiments? Can the authors provide an uncertainty analysis? While fine-tuning the FP model, how would the optimal QAT fraction for the final QAT phase shift? There might be a distribution difference between pre-training phase and fine-tuning phase.

---

> ### Author Response · Authors · 2025-11-19
> **Addressing weaknesses and questions**
>
> We thank the reviewer for the provided feedback. In this response we address the weaknesses and questions mentioned in the review.
>
> 1.
> > Experiments are conducted on a specific LLM architecture (llama2), and the scaling law parameters may require refitting for other model types.
> > Different types of models may exhibit varying degrees of tolerance to quantization, particularly those with inherent redundancy.
>
> Thank you for noting this. This is indeed an important point that we make sure to acknowledge in the paper ("limitations" paragraph of the conclusions).
> This issue is common in works on scaling laws [1,2] and natural to the problem itself.
> We do not currently see a practical way to mitigate this dependency, so acknowledging the limitation is the best we can do.
>
> 2.
> > While fine-tuning the FP model, how would the optimal QAT fraction for the final QAT phase shift?
> > There might be a distribution difference between the pre-training phase and the fine-tuning phase.
>
> As we understand, the question concerns post-pretraining stages (SFT, RL).
> It is true that we do not analyze how the optimal QAT fraction will change, as this would introduce a third stage into our experiments.
> Analyzing an additional post-pretraining stage (e.g., SFT or RL) would require a full three-stage scaling study with substantial extra compute and design choices;
> we therefore treat this as beyond the scope of the current work.
>
> However, we speculate on this topic in Appendix Q (Future Work, QAT Scaling Law for Multi-Stage Pretraining).
> We note that as the fine-tuning stage "constitutes a minor percentage of compute when compared to pretraining",
> one possible solution is to do fine-tuning entirely with QAT.
> All in all, this is an important direction of future work that is out of scope for our paper.
>
> 3.
> > How sensitive are the fitted parameters to the range of model sizes and token counts used in the experiments?
>
> We are working on this concern at the moment.
>
> **References**
>
> [1] Hoffmann, Jordan, et al. "Training compute-optimal large language models." arXiv preprint arXiv:2203.15556 (2022).
>
> [2] Bi, Xiao, et al. "Deepseek llm: Scaling open-source language models with longtermism." arXiv preprint arXiv:2401.02954 (2024).

---

> > ### Author Response · Authors · 2025-12-03
> > **Addressing weaknesses and questions**
> >
> > Please note that the newly uploaded revision addresses the question "How sensitive are the fitted parameters to the range of model sizes and token counts used in the experiments?" in Appendix Q (Uncertainty Analysis).
> >
> > Each parameter was analyzed for its influence on the quality of the loss model's fit, which was measured using two metrics: the $R^2$ value and the MAE of the optimal QAT fraction prediction. The analysis revealed that most parameters significantly impact the fit quality. The only exceptions are:
> > - $\alpha$: The constant irreducible loss term from the Chinchilla scaling law.
> > - $\kappa$ and $\theta$: The irreducible error terms associated with QAT.
> >
> > The statistical insignificance of these parameters for the optimal QAT fraction's MAE was expected. These parameters add a constant value, which affects the absolute loss value but not the relative position of the optimal QAT fraction (the argmin over a curve). As these parameters have an explicit conceptual meaning, we keep them in the loss model formula. Perhaps more important is that two other more complicated terms ("pure QAT penalty" and "FP / QAT interaction") are significant.

---

### Official Review · Reviewer_cjQW · 2025-11-01

**Soundness:** 3
**Presentation:** 3
**Contribution:** 3
**Rating:** 8
**Confidence:** 3

**Summary:**

This work is focused on QAT fine tuning which is done after full-precision (FP) phase pre-training.
QAT bit widths, and model sizes from 86.0M to2.2B to investigate how different QAT durations impact final performance.
They show that contrary to previous findings, the loss-optimal ratio of QAT to FP training increases with the total amount of compute.
This work derive a loss scaling law that predicts both optimal QAT ratios and final
model performance across different QAT/FP compute allocation strategies and
QAT bit widths.  Additionally, they propose a novel cooldown and QAT fusion approach that performs learning rate decay jointly with quantization aware training, eliminating redundant full-precision model updates and achieving significant compute savings.

**Strengths:**

Authors run multiple experiments for fitting generating scaling law into proposed function and demonstrated small prediction error. They showed that the optimal QAT fraction is not a fixed percentage but rather increases with the total compute budget, specifically with the tokens-per-parameterbyte statistic. This challenges the previous conclusion that 10% is universally optimal for QAT length relative to total training length.

They proposed a loss scaling law that captures the optimal QAT fraction phenomenon and
models the final expected loss of the FP and QAT.

Proposed a novel approach: cooldown & QAT fusion—a scheme where learning rate decay is performed jointly with quantization-aware training, eliminating redundant fullprecision updates and achieving better accuracy for the same token count.

**Weaknesses:**

This work is focused on a specific LLM architecture and data sets, and exact results may differ for different model types different QAT methods and data.
Only int quantization explored, it would be interesting to see fp4 and fp8 results.
Limited size LLM are explored.

**Questions:**

Do you reset or keep optimizer state, from pre-trained model, when apply QAT fine tuning?

---

> ### Author Response · Authors · 2025-11-19
> **Addressing weaknesses and questions**
>
> We thank the reviewer for the provided feedback and the high rating of our work.
> In this response we address the weaknesses and questions mentioned in the review.
>
> 1.
> > This work is focused on a specific LLM architecture and data sets, and exact results may differ for different model types, different QAT methods, and data.
>
> Thank you for noting this. This is indeed an important point that we make sure to acknowledge in the paper ("limitations" paragraph of the conclusions).
> This issue is common in works on scaling laws [1,2] and is natural to the problem itself.
> We do not currently see a practical way to mitigate this dependency, so acknowledging the limitation is the best we can do.
>
> 2.
> > Limited size LLMs are explored.
>
> It is true that we keep our model sizes relatively small (86M - 759M).
> However, we verify that our results are valid with scale increase by running experiments with a 2.2B model (Appendix K, "2.2B Model Optimal QAT Fraction Prediction").
> Indeed, the predicted optimal QAT fraction is experimentally optimal for most of the experiments or not far from optimality.
>
> 3.
> > Only int quantization explored, it would be interesting to see fp4 and fp8 results.
>
> We believe this to be an orthogonal direction to our work as lower-precision training (fp4, fp8, fp16) is not exclusive
> with int quantization.
> For example, int4 quantization can still be applied for a model trained in fp4 or fp8 precision for deployment
> on edge devices with limited support of fp4 or fp8.
>
>
> Perhaps, an interesting follow-up question is "how do scaling laws change when full-precision stage precision changes?"
> We do not conduct such experiments and highlight this question as a valuable future work direction.
> Based on previous works, we can speculate what results are expected (Appendix Q, "Future Work, Pretrain Precision & QAT Precision Interaction").
>
> 4.
> > Do you reset or keep the optimizer state from the pre-trained model when applying QAT fine-tuning?
>
> We reset the optimizer state when switching to QAT.
> As training dynamics change and new parameters are introduced, we do not see a way to re-use states.
> Neither have we encountered such approaches in the literature.
> This is also the default in prior QAT works we build on.
>
>
> **References**
>
> [1] Hoffmann, Jordan, et al. "Training compute-optimal large language models." arXiv preprint arXiv:2203.15556 (2022).
>
> [2] Bi, Xiao, et al. "Deepseek llm: Scaling open-source language models with longtermism." arXiv preprint arXiv:2401.02954 (2024).

---

### Official Review · Reviewer_Hsq3 · 2025-11-01

**Soundness:** 2
**Presentation:** 1
**Contribution:** 2
**Rating:** 2
**Confidence:** 4

**Summary:**

In this paper, the authors talk about the scaling law for quantization aware training as well as the impact of the fraction for quantization aware training compared to the full-precision pretraining. The topic seems to be interesting and the authors provide some experiments to demonstrate the results. However, the paper is very badly written and results presentation is very bad (e.g., the figures are very difficult to read and the conclusions made from the figures are difficult to follow or should be arguable).

**Strengths:**

- The topic and problem proposed seems to be interesting, and the final conclusion or results, if correct as the author claimed, might be useful. Especially, the conclusion about how to determine the optimal fraction for quantization aware training, and also the possibility to eliminate redundant full-precision training via novel learning rate scheduling, might be useful.
- Regarding the techinical contribution, the authors first study the optimal fraction allocated for QAT and its relationship with the tokens-per-parameter-byte. Afterwards, they propose the loss scaling law for optimal QAT fraction and model the final expected loss of the FP and QAT pipeline. They finally propose a novel approach of cooldown and QAT fusion, where learning rate cooldown in FP training and learning rate warmup in QAT are fused, and thus redundant full-precision updates can be eliminated.

**Weaknesses:**

- The writing and results presentation are not good. Figure1 is difficult to read.
- Improvement in Table 2 might not be strong enough.

**Questions:**

In Figure 1 left, the y-axis is QAT tokens-per-byte, and x-axis is Total tokens-per-byte. The author would like to demonstrate that the optimal QAT fraction increases with the full training tokens-per-parameter-byte (line 092 in the original paper). To demonstrate this, the y-axis should be the ratio between QAT tokens-per-byte to the total tokens-per-byte, instead of only list the two end ratios in the figure. Could the author provide the plot with ratio as the y-axis to prove the conclusion?

---

> ### Author Response · Authors · 2025-11-19
> **Addressing weaknesses and questions**
>
> We thank the reviewer for the provided feedback. In this response we address the weaknesses and questions mentioned in the review.
>
> 1.
> > Figure 1 is difficult to read.
>
> > In Figure 1 left, the y-axis is QAT tokens-per-byte, and the x-axis is Total tokens-per-byte. The author would like to demonstrate that the optimal QAT fraction increases with the full training tokens-per-parameter-byte (line 092 in the original paper).
> > To demonstrate this, the y-axis should be the ratio between QAT tokens-per-byte to the total tokens-per-byte, instead of only listing the two end ratios in the figure.
> > Could the author provide the plot with ratio as the y-axis to prove the conclusion?
>
> Thank you for raising this issue.
> We changed the y-axis of Figure 1 (left) to make the observed phenomenon clearer, and it is available in the updated submission.
>
> 2.
> > Improvement in Table 2 might not be strong enough.
>
> In lines 421-431 we note that perplexity improvements are hard to understand as improvements become smaller and smaller with the length of training.
> To address this issue, we calculate the improvement in terms of step difference. Most of the presented results show improvements around 10%,
> which we believe to be a strong efficiency improvement.
>
> Moreover, Cooldown & QAT fusion is not the main contribution of our work.
> Rather, it is a way to further improve QAT efficiency beyond optimal QAT fraction selection, contributing to the general plot of "Compute-Optimal QAT"
>
> 3.
> > The writing and results presentation are not good.
>
> We have updated some of our figures and added additional explanations (marked NEW or FIX). We will be happy to hear which sections' writing is unclear and what aspects of the results presentation are worth improving.

---

> ### Comment · Reviewer_Hsq3 · 2025-11-19
>
> Thanks the authors for the rebuttal effort.
>
> The updated Figure 1 seems to be more informative and illustrative. There seems to be some trend, but still a big fluctuations on the points (stars) compared to the dashed line.
>
> In Table 2, there can be several improvement. Firstly, it would be better if it is indicated that Unfused means baseline, and fused means something like "ours". Second, in the column name "Wasted tokens", it would be better to indicate somewhere that this is the metric you would like to focus mostly, and this metric should be as low as possible (e.g., with some downarrow on the side).
>
> Regarding the writing, the previous version of Figure 1 makes the reading difficult to understand the meaning of this work, but it is improved now. Section 3 might be better to be placed after the method or in the appendix, because it is not the most crucial part of the paper. Meanwhile, it is better to put the proposed method "Cooldown and QAT fusion" as the central part after the analysis of the key findings, i.e., the optimal fraction and the loss scaling law. Based on this, after section 2 of related work, the next section(s) should be something like "optimal QAT fraction" and "loss scaling law" to help introduce the method "cooldown and QAT fusion". The other parts like section 5 and 6 in the original submission should be better treated as ablation study to introduce some more related effects to help better understand the proposed method and the impact of other factors. In short, the method and key findings (with proper and good summary, and better with some takeaway or observations) should be collected together and mostly emphasized, immediately after the introduction and related work, to make the readers quickly understand the contribution of the work. Other non-critical analysis or study that are not the core of the study should better be placed after the key experimental results as ablation study. Emphasizing the key method, key findings, and main experimental results could help the presentation of the research. Hope this can be helpful.

---

> ### Author Response · Authors · 2025-11-19
>
> > In Table 2, there can be several improvement. Firstly, it would be better if it is indicated that Unfused means baseline, and fused means something like "ours". Second, in the column name "Wasted tokens", it would be better to indicate somewhere that this is the metric you would like to focus mostly, and this metric should be as low as possible (e.g., with some downarrow on the side).
>
> Thank you for these valuable suggestions. We agree that the notation in Table 2 can be improved to make it easier to understand, and we will incorporate the suggested changes in the next revision.
>
> Additionally, let us clarify the "wasted tokens" statistic. It should not be viewed as a standalone performance metric, but rather as a measure of the distance between the baseline (Unfused) and our method (Fused). In other words, higher "wasted tokens" indicate a larger improvement of our method over the baseline. Informally, this can be described as follows: if the same performance as the baseline can be achieved with fewer training tokens using our method, then all tokens that the baseline uses beyond this improved solution are counted as "wasted tokens".
>
> >  There seems to be some trend, but still a big fluctuations on the points (stars) compared to the dashed line.
>
> Please note that the dashed curve in Figure 1 (left) corresponds to the simple optimal-fraction fit introduced in the subsection “Predicting the Optimal QAT Fraction”. Its main purpose is to highlight the overall increasing trend rather than to perfectly interpolate all optimal points.
>
> Predictions obtained from the loss scaling law (subsection “Loss Scaling Law”) are more accurate, but they depend jointly on both QAT bit-width and tokens-per-parameter-byte. When these predictions are projected into 2D using only “Total tokens-per-byte” on the x-axis, the resulting curve becomes piecewise and visually less smooth.
>
> To avoid cluttering the main figure, we did not include these curves there. However, we agree that they can be informative, and we will add a plot in the appendix that shows loss–scaling-law-based optimal-fraction predictions separately for each bit-width.
>
> > Section 3 might be better to be placed after the method or in the appendix, because it is not the most crucial part of the paper.
>
> We will move Section 3 to Appendix A in order to improve the flow of the paper; this change will be reflected in the next revision.
>
> > In short, the method and key findings (with proper and good summary, and better with some takeaway or observations) should be collected together and mostly emphasized, immediately after the introduction and related work, to make the readers quickly understand the contribution of the work.
>
> We already summarize our key contributions in the introduction (in the paragraph beginning "The key contributions of this study are..."), so that readers are immediately aware of the main claims of the paper. We are not sure it is necessary to repeat this summary verbatim after the "Related Work" section, but as noted above, we will move Section 3 to the appendix to improve the overall flow.
>
> > Other non-critical analysis or study that are not the core of the study should better be placed after the key experimental results as ablation study. Emphasizing the key method, key findings, and main experimental results could help the presentation of the research. Hope this can be helpful.
>
> Our intended structure is as follows:
>
> - Scaling law fit and the phenomenon of the optimal QAT fraction increasing with compute (Section 4, "Optimal QAT Compute").
> - What can be inferred from the scaling law? (Sections 5, "When Does QAT Restore FP Model Accuracy?", and 6, "Parameter–Precision Trade-Off").
> - How can we improve QAT efficiency beyond the optimal QAT fraction? (Section 7, "Cooldown & QAT Fusion").
>
> To streamline this structure, we will move the paragraph "Quantifying Sub-Optimality" into its own subsection. We would like to emphasize that "Cooldown & QAT Fusion" is not the pinnacle of our work; rather, it is a method that further improves over the baseline once the optimal QAT fraction is chosen. Since the phenomenon of the optimal QAT fraction has not been explored in the literature before, and the "Quantifying Sub-Optimality" analysis shows that a sub-optimal QAT fraction can significantly harm final performance, we regard this as the main contribution of our work.
>
> Following your suggestion, we are considering making the high-level structure more explicit by organizing the core sections as:
>
> - "Optimal QAT Allocation" (with current subsections preserved).
> - "Loss Scaling Law Implications" (subsections: "Sub-Optimal QAT Fractions", "When Does QAT Match FP Accuracy?", and "Parameter–Precision Trade-Off").
> - "Beyond Fraction Allocation: Cooldown & QAT Fusion" (current "Cooldown & QAT Fusion" section).
>
> Please let us know what you think about this proposed structure and whether it would improve the flow and readability of the paper.

---

> > ### Comment · Reviewer_Hsq3 · 2025-11-19
> >
> > Thanks the authors for the further comments. By "observation" or "summary of method and key findings", it means some one sentence description of the key results for the current section. It is different from summary of contribution in the introduction section. Here is one good example for that: https://arxiv.org/abs/2201.03545. Note that at the end of several key subsections related to findings and methods, the authors use one sentence to summarize it. But it is not the key point. The main point is that, the current structure, from my humble point of view, is a little difficult to the readers to understand which parts are most crucial and important. There are four sections (4-7) all centering around the method and experiments, with both mixed together. It is not easy to follow which part are methods, which part are findings, and which part are experiments verifying the findings or the proposed methods.
> >
> > I do not believe a complete revision should be finished during the rebuttal period. It would require the reviewer to re-read the whole paper for once more time, and is costly. The best practice is to submit the fully revised version to another conference. Otherwise, if such revisions are repeated multiple turns, the reviewers might be exhausted by the reviewing process.

---

> > > ### Author Response · Authors · 2025-11-19
> > >
> > > > Thanks the authors for the further comments. By "observation" or "summary of method and key findings", it means some one sentence description of the key results for the current section. It is different from summary of contribution in the introduction section. Here is one good example for that: https://arxiv.org/abs/2201.03545. Note that at the end of several key subsections related to findings and methods, the authors use one sentence to summarize it.
> > >
> > > Thank you for the clarification and for pointing us to this example. We will study the suggested paper and, where space permits, add brief one‑sentence “takeaway” summaries at the end of key subsections to highlight the main result or observation of each part.
> > >
> > > > But it is not the key point. The main point is that, the current structure, from my humble point of view, is a little difficult to the readers to understand which parts are most crucial and important. There are four sections (4-7) all centering around the method and experiments, with both mixed together. It is not easy to follow which part are methods, which part are findings, and which part are experiments verifying the findings or the proposed methods.
> > >
> > > > I do not believe a complete revision should be finished during the rebuttal period. It would require the reviewer to re-read the whole paper for once more time, and is costly. The best practice is to submit the fully revised version to another conference. Otherwise, if such revisions are repeated multiple turns, the reviewers might be exhausted by the reviewing process.
> > >
> > > We fully understand your concern about reviewer workload and agree that a complete re‑write would not be appropriate during the rebuttal period. Our intention is not to overhaul the paper, but to slightly reorganize and rename sections so that the distinction between methods, key findings, and supporting experiments is clearer, while keeping the technical content and main results unchanged.
> > >
> > > Concretely, we plan to:
> > > - adjust section titles and grouping so that the core findings and the “Cooldown & QAT Fusion” method are more clearly separated from the scaling law analysis, and
> > > - add short takeaway sentences at the end of the most important subsections, as you suggest, to signal which results are central.
> > >
> > > These changes are meant to improve readability and make the structure more transparent, without requiring reviewers to re‑evaluate new experiments or substantially new content.

---

> > > > ### Author Response · Authors · 2025-11-19
> > > >
> > > > Dear reviewer, please take note of the recent submission (rebutal revision #2) that addresses the requested changes discussed in this thread up to this point.

---

### Author Response · Authors · 2025-11-19
**Revised submission**

We thank all reviewers for their feedback. We have now responded to all reviews and updated the manuscript accordingly. The revised submission incorporates the requested changes, and major edits are explicitly marked with "FIX" or "NEW". We are happy to engage in further discussion.

---

### Author Response · Authors · 2025-11-19
**Revised submission #2**

The revised submission incorporates the changes discussed with reviewer Hsq3. In addition, the implemented updates further address reviewer 1gxP’s concerns regarding the overall flow of the paper. Major edits are marked with "FIX" and "NEW" notes for reviewers’ convenience. This comment summarizes the main (mostly structural) changes:

1. The experimental context (previously Section 3) has been moved entirely to Appendix A so as not to disrupt the main narrative flow.
2. "Takeaway" boxes with brief section summaries have been added to improve readability and highlight the key messages of each part.
3. The main sections and subsections have been reorganized into three clearer parts to make the overall structure easier to follow:
   - Optimal QAT Compute Allocation
     - Predicting the Optimal QAT Fraction
     - Loss Scaling Law
   - Loss Scaling Law Implications
     - Evaluating Sub-Optimal QAT Fraction
     - When Does QAT Match FP Accuracy?
     - Parameter–Precision Trade-Off
   - Beyond QAT Compute Fraction: Cooldown & QAT Fusion

We hope these changes improve the flow of the paper and clarify how each section relates to the others.

As the revisions are primarily structural and do not alter the main technical content, we hope reviewers do not feel overburdened by this additional version.

---

### Author Response · Authors · 2025-12-03
**Rebuttal Summary**

We thank all the reviewers for taking the time to review our work and for their valuable feedback. We have uploaded an improved version of the paper that addresses the concerns raised by the reviewers as well as minor typos and grammatical errors. Below, we summarize broad improvement areas and the corresponding changes made in the paper for the AC's convenience.


1. **Paper Flow and Structure**

Several reviewers (1gxP, Hsq3) noted that the structure of the paper may be hard to follow. To address this, we restructured sections to clearly separate the main part of the paper into three sections (Optimal QAT Compute Allocation, Loss Scaling Law Implications, Beyond QAT Compute Fraction: QAT & Cooldown Fusion). We also moved the old Section 3 (Experimental Setup) to Appendix A to improve the flow of the paper.

Apart from making structural changes, we have added short takeaway messages to all the main sections of the paper. We hope this will further facilitate understanding of the paper.

2. **Paper Limitations**

Both reviewers w7Wt and cjQW highlighted possible limitations of the paper, specifically the specific LLM architecture and dataset used. This is indeed an important point that we make sure to acknowledge in the paper ("limitations" paragraph of the conclusions). This issue is common in works on scaling laws and is natural to the problem itself. Given the extensive cost of running experiments at the scale of this work, we do not currently see a practical way to mitigate this dependency, so acknowledging the limitation is the best we can do.

Still, we verify that the results of our work are not dataset- or hyperparameter-induced in Appendix J (Dataset and Hyperparameter Impact) and that they are scalable to larger model sizes in Appendix K (2.2B Model Optimal QAT Fraction Prediction).


3. **Scaling Law Fit Uncertainty and Significance Analysis**

As requested by reviewer w7Wt and hinted at by reviewer 1gxP ("[...] What is the pure QAT penalty, and why does it take that form? Why does the FP/QAT interaction take this form? Are these grounded in theory or chosen for fitting reasons? Did you do any ablation studies on the form?"), we added a new section to the appendix that performs uncertainty analysis of the fit and determines which parameters are statistically significant.


4. **Minor Presentation Issues**

Reviewers 1gxP and Hsq3 proposed valuable advice on improving figures, tables, and equations. Additionally, we have added more detailed explanations where requested.

---

Apart from the broad areas described above, minor changes were made to address specific comments from the reviewers. We made sure to respond to each question and weakness raised by the reviewers in each specific response thread.

---

### Meta-Review · Area_Chair_6QKa · 2025-12-31

**Summary:**

This paper studies how to allocate compute between FP pretraining and a subsequent QAT stage.

Through a large experimental sweep across model sizes, token budgets, and QAT bit-widths, the authors report (i) the loss-optimal QAT fraction increases with total compute (contrary to fixed-percentage heuristics), (ii) a fitted loss scaling law that predicts both the optimal QAT fraction and final performance under different FP/QAT allocations and bit-widths, and (iii) an additional “cooldown & QAT fusion” schedule aimed at reducing redundant FP updates by merging LR decay with QAT.

I recommanded Accept as poster. Since i belive the initial low score (2) should have lean at least to borderline reject, as the concerns raised are minor. Other reviewers are storngly support paper acceptance and I beliave that the paper provides vital insgihts which will impact the field of efficent training.

**Reviewer Concerns:**

* **Presentation / narrative clarity**: one reviewer found the paper “badly written” initially (rating 2), and another found it long with a central narrative that was hard to follow, with key definitions (e.g., “wasted tokens”) under-explained, however, I don't find it as something that can't be slighly improved in the camera ready version.

* **Generality limits**: experiments appear tied to a specific architecture/data regime and integer quantization; reviewers note parameters may need refitting for other model families or quantization/training variants.

* **Model-form justification**: a reviewer asked for grounding/ablation on the chosen functional forms (pure QAT penalty and FP/QAT interaction). Authors state the forms are empirically chosen and that simpler forms did not fit well; still, this remains largely empirical rather than theoretically motivated.

**Reviewer Scores:**

## Reviewer Hsq3: 2 → 4
Why: Their main concerns were readability/structure and figure/table clarity rather than disputing the core empirical claims. They already acknowledged Figure 1 improved and then focused on presentation suggestions. With full discussion and the authors’ structural revisions + clarified metrics, I expect a move from “reject” to “borderline reject.”

## Reviewer cjQW: 8 → 8
Why: Already strongly positive on soundness/contribution and had a concrete technical question (optimizer state) that the authors answered clearly. No remaining major concern that would drive a score change.

## Reviewer w7Wt: 6 → 8
Why: Primary concern was sensitivity/uncertainty of the fitted parameters and generality. The authors added an uncertainty/significance analysis in a later revision and addressed the question directly. Full discussion would likely increase confidence, but residual scope limits keep it from jumping to the very top.

## Reviewer 1gxP: 8 → 8
Why: Strong accept with mostly clarity/packaging issues and requests for definitions/motivation. Authors added formalizations for “wasted tokens,” clarified low-token divergence, and acknowledged the empirical nature of the model form. Full discussion likely reinforces acceptance without changing the headline score.

---

### Decision · Program_Chairs · 2026-01-26

Accept (Poster)